# Human influenza virus infection elicits distinct patterns of monocyte and dendritic cell mobilization in blood and the nasopharynx

Sindhu Vangeti[1], Sara Falck-Jones[1], Meng Yu[1], Björn Österberg[1], Sang Liu[1], Muhammad Asghar[2,3], Klara Sondén[2,3], Clare Paterson[4], Penn Whitley[5], Jan Albert[6,7], Niclas Johansson[2,3], Anna Färnert[2,3], Anna Smed-Sörensen[1]*

[1]Division of Immunology and Allergy, Department of Medicine Solna, Karolinska Institutet, Karolinska University Hospital, Stockholm, Sweden; [2]Division of Infectious Diseases, Department of Medicine Solna, Karolinska Institutet, Stockholm, Sweden; [3]Department of Infectious Diseases, Karolinska University Hospital, Stockholm, Sweden; [4]SomaLogic Operating Co., Inc,, Boulder, United States; [5]Boulder Bioconsulting Inc, Boulder, United States; [6]Department of Microbiology, Tumor and Cell Biology, Karolinska Institutet, Stockholm, Sweden; [7]Department of Clinical Microbiology, Karolinska University Hospital, Stockholm, Sweden

*For correspondence:
anna.smed.sorensen@ki.se

**Abstract** During respiratory viral infections, the precise roles of monocytes and dendritic cells (DCs) in the nasopharynx in limiting infection and influencing disease severity are incompletely described. We studied circulating and nasopharyngeal monocytes and DCs in healthy controls (HCs) and in patients with mild to moderate infections (primarily influenza A virus [IAV]). As compared to HCs, patients with acute IAV infection displayed reduced DC but increased intermediate monocytes frequencies in blood, and an accumulation of most monocyte and DC subsets in the nasopharynx. IAV patients had more mature monocytes and DCs in the nasopharynx, and higher levels of TNFα, IL-6, and IFNα in plasma and the nasopharynx than HCs. In blood, monocytes were the most frequent cellular source of TNFα during IAV infection and remained responsive to additional stimulation with TLR7/8L. Immune responses in older patients skewed towards increased monocyte frequencies rather than DCs, suggesting a contributory role for monocytes in disease severity. In patients with other respiratory virus infections, we observed changes in monocyte and DC frequencies in the nasopharynx distinct from IAV patients, while differences in blood were more similar across infection groups. Using SomaScan, a high-throughput aptamer-based assay to study proteomic changes between patients and HCs, we found differential expression of innate immunity-related proteins in plasma and nasopharyngeal secretions of IAV and SARS-CoV-2 patients. Together, our findings demonstrate tissue-specific and pathogen-specific patterns of monocyte and DC function during human respiratory viral infections and highlight the importance of comparative investigations in blood and the nasopharynx.

## Editor's evaluation

This study presents a valuable evaluation of the distribution of monocytes and dendritic cells in the blood and nasopharyngeal aspirates of patients with mild respiratory tract infections. They report solid differences between monocytes and dendritic cells and variation with patient age, showing that during influenza A virus infection, the classical and intermediate monocyte numbers increased

both, in blood and nasopharyngeal aspirates while DCs increased in the nasopharyngeal aspirates only. The work will be of broad interest to immunologists, lung biologists and infection disease community.

## Introduction

Respiratory viral infections cause significant global disease burden with influenza, or flu, being responsible for a significant portion. An estimated 1 billion cases of influenza occur annually resulting in approximately 3–5 million severe cases and 290,000–650,000 deaths (*WHO, 2018*). In addition to seasonal epidemics caused by both influenza A or B virus (IAV and IBV, respectively), IAV can also cause pandemics. The majority of infections remain asymptomatic or develop mild to moderate respiratory disease, characterized by fever, nasal congestion, cough, and muscle aches. Severe disease mainly affects infants, pregnant women, and the elderly or immunocompromised, but can also occur in otherwise healthy individuals (*Cole et al., 2017*; *Oshansky et al., 2014*). The determinants of disease severity are still incompletely understood but may include properties of the virus, environmental factors, genetics, and immune responses of the patient (*Kuiken and Taubenberger, 2008*; *Kuiken et al., 2010*). Other agents causing an influenza-like illness (ILI) during influenza season include respiratory syncytial virus (RSV) and seasonal coronaviruses (OC43, HKU1, 229E, and NL63). Similar to IAV, coronaviruses are capable of causing pandemics, most notably the ongoing coronavirus disease 2019 (COVID-19) caused by the severe acute respiratory syndrome coronavirus 2 (SARS-CoV-2) (*Zhou et al., 2020*).

IAV is primarily transmitted via inhalation of virus-containing aerosols or droplets, and mainly targets respiratory epithelial cells (*Tellier, 2006*; *Sanders et al., 2011*), with the nasopharynx being the initial site of virus replication. IAV generally remains localized to the airways, despite signs of systemic inflammation (*Stramer et al., 2012*). At the site of infection, resident innate immune cells including monocytes and dendritic cells (DCs) rapidly respond to the presence of virus by secreting cytokines, interferons, and chemokines to limit viral spread and recruit immune cells (*Sakabe et al., 2011*; *Wang et al., 2012*). Monocytes and DCs shape the specificity and strength of the subsequent adaptive responses (*Kwissa et al., 2014*; *Spörri and Reis e Sousa, 2005*). In blood, three subsets of monocytes are found: the CD14+CD16– classical monocytes (CMs), the most frequent subset at steady state; the terminally differentiated CD14–CD16+ nonclassical monocytes (NCMs); and CD14+CD16+ intermediate monocytes (IMs) that represent CMs transitioning into NCMs (*Boyette et al., 2017*; *Jakubzick et al., 2013*; *Patel et al., 2017*). Blood IMs are readily identifiable at steady state and expand rapidly in response to inflammation, infection, or vaccination (*Oshansky et al., 2014*; *Kwissa et al., 2014*; *Fingerle et al., 1993*). In addition, monocytes extravasate to tissue where they play an important role in innate immune protection. Monocytes also secrete TNFα, a major regulator of innate immune function that is central to the cytokine storm associated with IAV infection (*Tisoncik et al., 2012*). Conventionally, DCs were grouped as the CD1c+ myeloid DCs (MDCs) that excel at activating naïve T cells *Hémont et al., 2013*; the CD141+ MDCs that can efficiently can cross-present antigens via MHC-I *Jongbloed et al., 2010*; and the CD123+ plasmacytoid DCs (PDCs) that mediate type I IFN responses (*Cella et al., 1999*). More recent single-cell sequencing and unbiased clustering revealed six subtypes of DCs and four subtypes of monocytes (*Villani et al., 2017*). In this manuscript, we focus on cDC1 (previously CD141+ MDCs) and cDC2 (previously CD1c+ MDCs). We also speculate the roles for DC3s (inflammatory CD14-expressing CD1c+ MDCs, which seemingly arise in a GM-CSF-dependent manner) (*Bourdely et al., 2020*) and pre-DCs (AXL+SIGLEC6+CD11c$^{low}$CD123+DCs that do not produce IFNα upon TLR9 stimulation but produce IL-12p70 at levels comparable to cDCs) (*Villani et al., 2017*; *See et al., 2017*). Monocytes and DCs vary in distribution and function, depending on the anatomical compartment (*Baharom et al., 2016*; *Alcántara-Hernández et al., 2017*). Moreover, monocytes and DCs are susceptible to IAV infection in vitro, and the cytopathic nature of the virus may impair their antigen processing and presenting functions (*Smed-Sörensen et al., 2012*; *Diao et al., 2014*; *Baharom et al., 2015*), delaying recovery and normalization of immune cell distribution and function (*Jochems et al., 2018*; *Lichtner et al., 2011*).

Studies have shown that monocytes and DCs are recruited to the nasopharynx following infection with 2009 H1N1pdm IAV strains (*Oshansky et al., 2014*), and in individuals hospitalized with severe influenza infections (*Gill et al., 2008*; *Gill et al., 2005*). Disease severity in hospitalized patients has

been shown to correlate with (1) monocyte recruitment and increased levels of MCP3, IFNα-2, and IL-10 in the nasal compartment (*Oshansky et al., 2014*) and (2) strong TNF-producing monocytic responses in blood (*Cole et al., 2017*) and inflammatory, neutrophil-dominant patterns (*Dunning et al., 2018*). Despite accounting for a comparatively greater burden of disease, immune responses during mild seasonal influenza infections remain less studied. Therefore, the roles played by monocytes and DCs in contributing to or mitigating mild influenza disease are largely unknown. Additionally, while the response of blood monocytes and DCs to IAV has been studied well in vitro and in animal models (*Smed-Sörensen et al., 2012*; *Diao et al., 2014*; *Jochems et al., 2018*; *Vangeti et al., 2019*; *McGill et al., 2008*; *Mount and Belz, 2010*; *GeurtsvanKessel et al., 2008*), few studies compare responses between blood and the nasopharynx in human infections (*Dunning et al., 2018*; *Vangeti et al., 2019*; *Segura et al., 2013*). Immune cell behaviour in the nasopharynx during IBV and RSV infections has not been studied in great detail but evidence of DC mobilization to the nasal cavity has been reported (*Gill et al., 2008*). Studies on immune responses to mild SARS-CoV-2 infection have also primarily focused on blood and rarely the upper airways (*Mick et al., 2020*).

Here, we determined monocyte and DC subset distribution, maturation and function in both blood and, for the first time, the nasopharynx, in patients with mild to moderate seasonal influenza and influenza-like infections. The methods described in this study allowed us to investigate airway immunity in a larger cohort of patients with SARS-CoV-2 infection (*Falck-Jones et al., 2021*; *Cagigi et al., 2021*), showing that methods to study the immune responses in the nasopharynx during acute disease are essential tools as we face the possibility of future pandemics. Comparing the dynamics of systemic and nasopharyngeal immune function will add to our understanding of the roles of monocytes and DCs in shaping the nature and magnitude of inflammation and subsequently disease severity during respiratory viral infections.

## Results

### Study subject characteristics

During five consecutive influenza seasons (2016–2020), 121 adults with symptoms of ILI were included in the study. Blood, nasal swabs, and nasopharyngeal aspirates were collected (*Figure 1A*). IAV infection was confirmed by PCR in 70 patients while 51 patients had other infections (IBV: 10, RSV: 6, SARS-CoV-2: 35) despite presenting with similar symptoms (*Figure 1B*), consistent with inclusion based on ILI. No annual pattern of viral etiology was observed across the 5 years, except for SARS-CoV-2 which emerged in late 2019 and was only seen in patients from 2020 (*Figure 1C*). The severity of disease in patients was categorized as 'mild' or 'moderate' (detailed description in Methods). Of the 121 patients, 35 had moderately severe disease and 64 were hospitalized (*Table 1*). Of the SARS-CoV-2 patients, 15 were characterized as mild and 20 had moderate disease. Twenty healthy controls (HCs) were included and sampled identically.

IAV patients had a median age of 59 (range: 20–98 years) and sought medical attention after a median of 4 days following onset of symptoms (IQR: 2–5) (*Table 1*). HCs had a median age of 44 (range: 22–59 years). IAV patients had a median Charlson comorbidity index (CCI) of 1.5 (IQR: 0–4), and 36 IAV patients had at least one underlying comorbidity (chronic heart/lung diseases, reduced lung function, kidney insufficiency, diabetes mellitus, asplenia/hyposplenia, and malignancies).

### Human IAV infection is characterized by an influx of CD11c+ cells into the nasopharynx

Blood from IAV patients yielded significantly fewer PBMCs/mL compared to HCs (*Figure 1D*). In contrast, threefold higher cell numbers were recovered from the nasopharynx of IAV patients compared to HCs (median 0.77 vs. 0.25 × $10^6$ cells). In fact, 43% of IAV patients had more than 1×$10^6$ cells recovered from their NPA sample. Viability of PBMCs and NPA cells was variable across individuals, with no statistically significant differences between patient or HC groups (*Figure 1E*). We determined the immune cell distribution in blood and the nasopharynx by flow cytometry (*Figure 1—figure supplement 1*) on matched PBMC and NPA samples with a minimum $10^5$ cells and ≥70% viability (n=22 IAV patients and n=16 HCs) to obtain high-quality data (stained with identical panels and clones of antibodies). The frequencies of live CD45+ immune cells were increased in the NPA of patients as compared to HCs but remained similar in blood between the groups (data not shown).

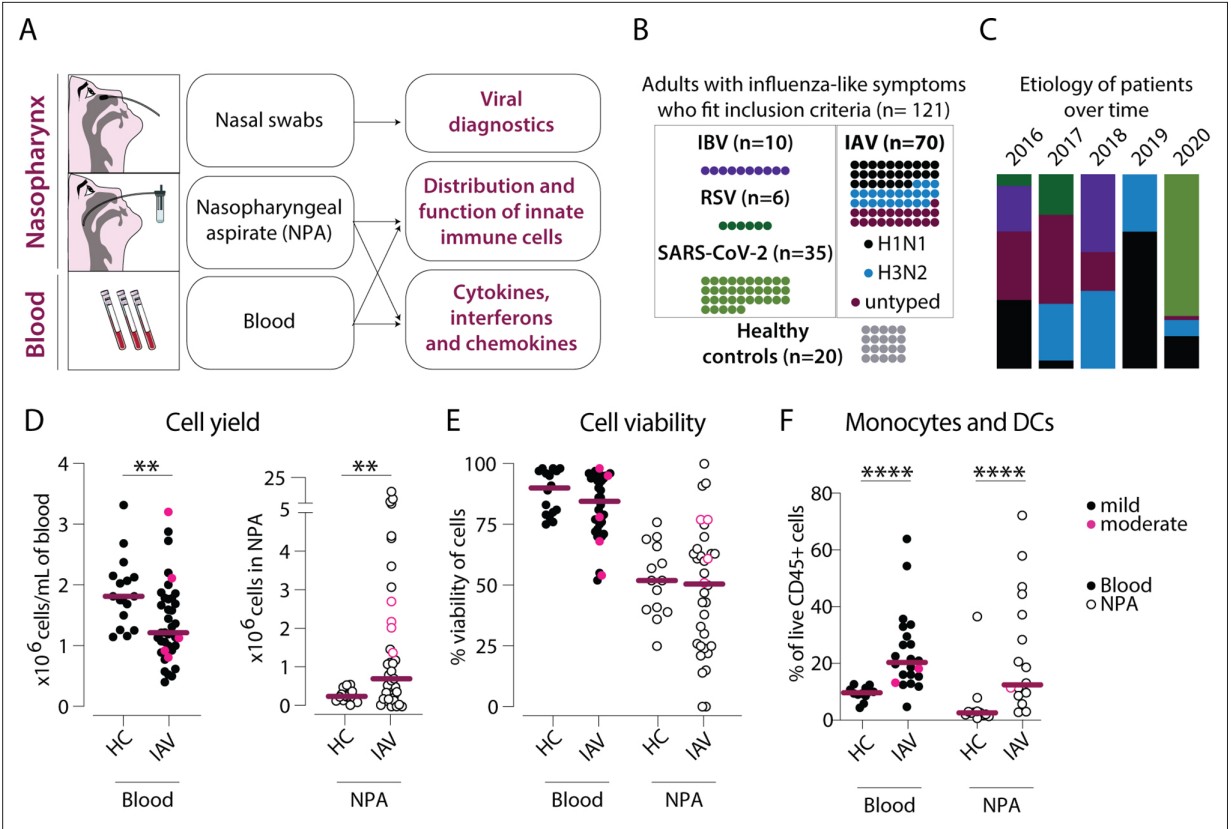

**Figure 1.** Cellular infiltration into the nasopharynx during influenza A virus (IAV) infection is largely due to accumulation of lineage negative HLA-DR+ cells. (**A**) Nasal swabs, nasopharyngeal aspirates (NPA), and peripheral blood samples were collected from patients with acute symptoms of influenza-like symptoms and during their convalescence as well as from healthy controls (HCs). (**B**) Forty of 64 patients with influenza-like symptoms were confirmed to be infected with IAV by PCR and were included in the study. Further, 27 of the IAV patients were infected with H1N1, and 22 with H3N2. A variety of analyses were performed on acute (n=121) and convalescent (n=11) samples from the IAV patients and HC (n=20). (**C**) Graph depicts the relative distribution of IAV, IBV, respiratory syncytial virus (RSV), and severe acute respiratory syndrome coronavirus 2 (SARS-CoV-2) cases during each year of inclusion to illustrate the shifting trends in circulating respiratory viral illnesses over time. (**D–F**) Scatter plots show data from individual subjects and lines indicate median values. IAV patients with mild disease, as defined by peak respiratory sequential organ failure assessment (SOFA) or modified SOFA (mSOFA) score of 1 or 2 are indicated in black and those with moderate disease (mSOFA score of 3 or 4) are indicated in pink. (**D**) ×10⁶ PBMCs (per mL blood, filled circles) and ×10⁶ total NPA cells (open circles) obtained from IAV patients and HCs. (**E**) Cell viability of PBMCs and NPA from patients and HCs was assessed using Trypan Blue exclusion staining and manual counting. (**F**) Frequency of lineage (CD3, CD19, CD20, CD56, CD66abce) negative HLA-DR+ cells (monocytes and myeloid dendritic cells) of live CD45+ cells in blood and NPA from HCs (n=16) and IAV patients (n=22). Differences between IAV patients and HCs were assessed using Mann-Whitney test and considered significant at p<0.05 (**p<0.01, ****p<0.0001).

The online version of this article includes the following source data and figure supplement(s) for figure 1:

**Source data 1.** Related to *Figure 1D* – blood.

**Source data 2.** Related to *Figure 1D* – nasopharyngeal aspirates (NPA).

**Source data 3.** Related to *Figure 1E* – blood.

**Source data 4.** Related to *Figure 1E* – nasopharyngeal aspirates (NPA).

**Source data 5.** Related to *Figure 1F* – blood.

**Source data 6.** Related to *Figure 1F* nasopharyngeal aspirates (NPA).

**Figure supplement 1.** Gating strategy for identification of monocytes and dendritic cells from PBMCs and nasopharyngeal aspirates (NPA).

Among the immune cells, we found significantly higher frequencies of lineage (CD3, CD19, CD20, CD56, CD66abce) negative, HLA-DR+ cells, the compartment where monocytes and MDCs can be identified, in both blood (p<0.0001) and the NPA (p<0.01) of IAV patients as compared to HCs (*Figure 1E*). Therefore, our data show that acute IAV infection results in an influx of monocytes and DCs to the nasopharynx.

**Table 1.** Patient and control characteristics.

| Cohort | IAV | IBV | RSV | SARS-CoV-2 | Healthy controls | p-Value* |
|---|---|---|---|---|---|---|
| n | 70 | 10 | 6 | 35 | 20 | |
| Age, median (range) | 59 (20–98) | 54 (25–89) | 63 (32–88) | 52 (26–76) | 44 (22–59) | 0.009 |
| Male gender, n (%) | 29 (41) | 6 (60) | 2 (33) | 18 (51) | 12 (60) | 0.5 |
| Onset to inclusion, days, median [IQR] | 4 [2–5] | 5 [4–6] | 6 [4–7] | 12 [10–22] | – | <0.001 |
| Hospital admission, n (%) | 34 (49) | 2 (20) | 4 (67) | 24 (69) | – | 0.031 |
| Comorbidities | | | | | | |
| CCI, median [IQR] | 1.5 [0.0–4.0] | 0.0 [0.0–3.0] | 2.50 [2.0–3.0] | 1.0 [0.0–3.0] | – | 0.2 |
| BMI, median [IQR] | 25.9 [23.0–31.6] | 26.3 [23.4–29.4] | 26.0 [22.6–29.9] | 27.0 [25.4–30.5] | – | 0.8 |
| Hypertension, n (%) | 20 (29) | 1 (10) | 2 (33) | 9 (26) | – | 0.7 |
| Diabetes, n (%) | 7 (10) | 2 (20) | 2 (33) | 9 (26) | – | 0.073 |
| Current smoker, n (%) | 11 (17) | 1 (11) | 0 | 2 (5.9) | – | 0.4 |
| Laboratory analyses | | | | | | |
| CRP (mg/L), median [IQR] | 40 [24–67] | 34 [9–39] | 44 [33–62] | 102 [24–182] | 0 (0–0) | <0.001 |
| WBC (×$10^9$/L), median (IQR) | 6.4 [5.0–8.3] | 5.8 [5.3–9.0] | 8.9 [NA] | 6.7 [4.5–7.9] | 6.0 [5.0–6.8] | 0.3 |
| Lymphocytes (×$10^9$/L), median [IQR] | 1.0 [0.7–1.3] | 0.8 [0.7–0.9] | 1.1 [NA] | 1.20 [0.8–1.7] | 1.7 [1.5–2.0] | 0.048 |
| Neutrophils (×$10^9$/L), median [IQR] | 4.7 [2.7–6.3] | 4.6 [3.3–6.8] | 5.6 [NA] | 4.4 [2.7–6.6] | 3.2 [3.0–3.8] | 0.9 |
| Monocytes (×$10^9$/L), median [IQR] | 0.7 [0.5–0.8] | 0.5 [0.5–0.6] | 0.8 [NA] | 0.4 [0.3–0.6] | 0.5 [0.4–0.5] | 0.022 |
| Ct value, median [IQR] | 25.1 [22.1–31.1] | 27.2 [22.4–31.8] | 30.1 [25.8–35.9] | 27.1 [23.0–29.2] | – | 0.3 |
| Treatment | | | | | | |
| Tamiflu prescribed, n (%) | 35 (54) | 0 | 0 | 0 | – | – |
| Antibiotics prescribed, n (%) | 21 (33) | 2 (22) | 3 (50) | NA | – | 0.5 |
| Peak severity score | | | | | | |
| Mild disease, n (%) | 57 (81) | 10 (100) | 4 (67) | 15 (43) | – | <0.001 |
| Moderate disease, n (%) | 13 (19) | 0 | 2 (33) | 20 (57) | – | <0.001 |

*Statistical significance was determined by Kruskal-Wallis rank sum test; Fisher's exact test. CCI: Charlson comorbidity index. BMI: body mass index. CRP: C-reactive protein. WBC: white blood cells. Ct: cycle threshold. Normal range: BMI: 18.5–24.9, CRP <3 mg/L, WBC 3.5×$10^9$/L to 8.8×$10^9$/L, lymphocytes 1.1×$10^9$/L to 3.5×$10^9$/L, neutrophils 1.6×$10^9$/L to 5.9×$10^9$/L, monocytes 0.2×$10^9$/L to 0.8×$10^9$/L.

## Increased frequencies of CM and IM in the nasopharynx during IAV infection

To identify which monocyte and DC subsets contributed to the changes observed in the myeloid cell compartment during IAV infection, we compared the distribution of the different cell subsets. As expected, CMs were the most frequent monocytes in blood in both patients and HCs, and remained comparable. However, in the nasopharynx of IAV patients as compared to HCs, blood CM frequencies were significantly increased (*Figure 2A*). Strikingly, IM frequencies were significantly elevated, in both blood and NPA of IAV patients (*Figure 2B*), while blood NCM appeared to be lower in patients compared to HCs (*Figure 2C*). In HCs, the nasopharynx was virtually devoid of DCs– cDC2s, cDC1s, PDCs, and CD14+CD1c+ monocyte-derived DCs (mo-DCs) were only identified in a subset of HCs

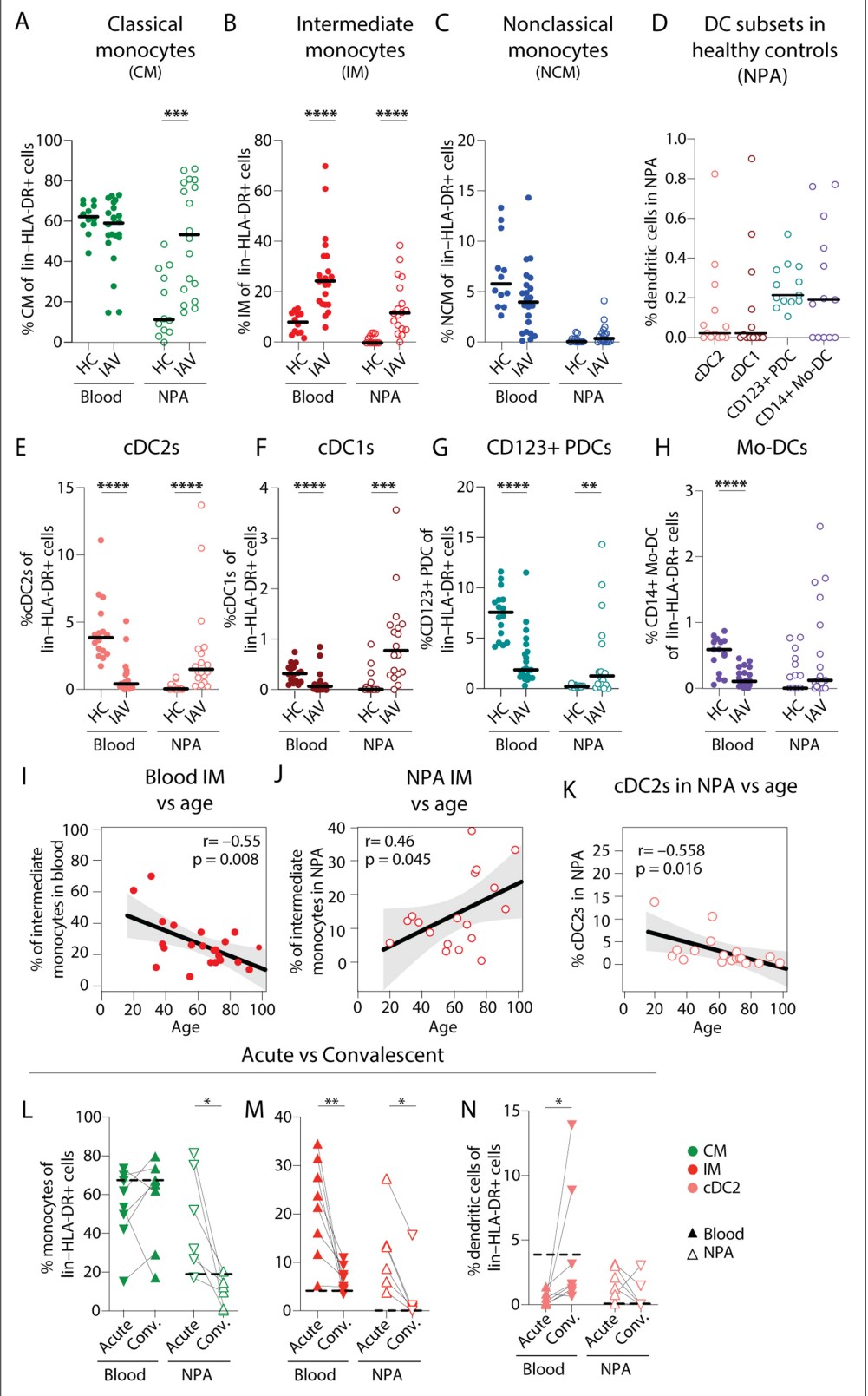

**Figure 2.** Intermediate monocyte and dendritic cell frequencies are significantly altered in blood and the nasopharynx from influenza A virus (IAV) patients compared to healthy controls. (**A–C**) Scatter plots show frequencies of (**A**) CD14+CD16–CD1c– classical monocytes (CMs), (**B**) CD14+CD16+ intermediate monocytes (IMs), and (**C**) CD14–CD16+ nonclassical monocytes (NCMs) in PBMCs and nasopharyngeal aspirates (NPA) from

*Figure 2 continued on next page*

*Figure 2 continued*

healthy controls (HCs) (n=12) and IAV patients (n=22). (**D**) Scatter plots shows the frequencies of cDC2s (coral), cDC1s (maroon), CD123+ plasmacytoid dendritic cells (PDCs) (teal) and CD14+ monocyte-derived DCs (mo-DCs) (purple) in the NPA from HCs (n=16). (**E–H**) Graphs show frequencies of (**E**) cDC2s, (**F**) cDC1s, (**G**) CD123+ PDCs, and (**H**) mo-DCs expressed as a frequency of lin–HLA-DR+ cells in PBMCs and NPA from HCs (n=12) and IAV patients (n=22). Lines (**A–H**) indicate median values. Differences between IAV patients and HCs were assessed using Mann-Whitney test and considered significant at p<0.05 (**p<0.01, ***p<0.001, ****p<0.0001). (**I–K**) Scatter plots and line of fit display bivariate linear regression analysis between age of IAV patients and IM frequency in (**I**) blood and (**J**) NPA; and (**K**) between age and frequency of cDC2s in blood. The shaded area represents the 95% confidence region for the fitted line. R represents Spearman $\rho$ and differences were considered significant at p<0.05. (**L**) Graphs depict frequencies of classical monocytes (CMs) (green), intermediate monocytes (IMs) (red), and cDC2s (coral) in blood (n=8) and the NPA (n=6) in the acute (upward triangles) and convalescent phase (downward triangles) in IAV patients. Dashed lines depict median frequency values from HCs in blood and NPA. Differences between acute and convalescent phase values were assessed using Wilcoxon matched-pairs signed rank test and considered significant at p<0.05 (*p<0.05 and **p<0.01).

The online version of this article includes the following source data and figure supplement(s) for figure 2:

**Source data 1.** Related to *Figure 2A* – blood.

**Source data 2.** Related to *Figure 2A* – nasopharyngeal aspirates (NPA).

**Source data 3.** Related to *Figure 2B* – blood.

**Source data 4.** Related to *Figure 2B* – nasopharyngeal aspirates (NPA).

**Source data 5.** Related to *Figure 2C* – blood.

**Source data 6.** Related to *Figure 2C* – nasopharyngeal aspirates (NPA).

**Source data 7.** Related to *Figure 2D* – nasopharyngeal aspirates (NPA).

**Source data 8.** Related to *Figure 2E* – blood.

**Source data 9.** Related to *Figure 2E* – nasopharyngeal aspirates (NPA).

**Source data 10.** Related to *Figure 2F* – blood.

**Source data 11.** Related to *Figure 2F* – nasopharyngeal aspirates (NPA).

**Source data 12.** Related to *Figure 2G* – blood.

**Source data 13.** Related to *Figure 2G* – nasopharyngeal aspirates (NPA).

**Source data 14.** Related to *Figure 2H* – blood.

**Source data 15.** Related to *Figure 2H* – nasopharyngeal aspirates (NPA).

**Source data 16.** Related to *Figure 2I*.

**Source data 17.** Related to *Figure 2J*.

**Source data 18.** Related to *Figure 2K*.

**Source data 19.** Related to *Figure 2L*.

**Source data 20.** Related to *Figure 2M*.

**Source data 21.** Related to *Figure 2N*.

**Figure supplement 1.** Gating strategy for identification of CD11c^low cDC1s from PBMCs and nasopharyngeal aspirates (NPA).

(*Figure 1—figure supplement 1* and *Figure 2D*). In contrast, during acute IAV infection, all DC subsets were significantly decreased in the blood of patients (*Figure 2E–H*). Meanwhile in the nasopharynx, cDC2, cDC1, and PDC frequencies were significantly higher in IAV patients compared to HCs (*Figure 2E–H*). Additionally, we looked from the rare cDC1s with low CD11c expression (*Figure 2—figure supplement 1*). Despite very low numbers, the same trend of significant reduction in blood and increased presence in the nasopharynx was observed for these cells as well.

We also compared the frequency of IMs in blood and nasopharynx with the age of IAV patients and found a negative correlation in blood (R=−0.55, p=0.0008) (*Figure 2H*) but a positive correlation in the nasopharynx (R=0.46, p=0.045) (*Figure 2I*). In HCs, age and IM frequencies were not significantly correlated in blood or NPA (data not shown). In contrast to IMs, we found an inverse correlation between age and the frequency of cDC2s in NPA (R=−0.558, p=0.016) in IAV patients (*Figure 2K*).

A subset of IAV patients (n=11) returned for sampling during convalescence (≥4 weeks after initial sampling). We observed that frequencies of CMs (in blood), IMs (blood and NPA), and cDC2s (blood) in convalescent individuals were reduced and closer to values seen in HCs (*Figure 2L*). Collectively, we found that the increased immune cell presence in the nasopharynx during acute IAV infection could be, to a large extent, attributed to increased frequencies of IMs as well as CMs which normalized during convalescence. Acute IAV infection resulted in altered monocyte distribution, in particular at the site of infection and was more pronounced in older patients.

## Monocytes and DCs recruited to the human nasopharynx during IAV infection are mature

We next analysed the maturation status of DCs and monocytes in blood and nasopharynx samples from IAV patients and HCs (*Figure 3A–D*). Cells from HCs had low and comparable levels of surface HLA-DR in both blood and the nasopharynx (*Figure 3A*). In contrast, in IAV patients, monocytes and DCs in the nasopharynx expressed higher levels of HLA-DR than those in blood (*Figure 3A and C*). We also found that in IAV patients, nasopharyngeal cDC2s, and cDC1s expressed more CD86 than cells in blood (*Figure 3D*), while nasopharyngeal PDCs showed significant upregulation of CD83 during IAV infection as compared to blood PDCs (*Figure 3E*). When we compared maturation of CM and cDCs (i.e., CD86 expression) with viral RNA load (i.e., cycle threshold [Ct] values – number of cycles required to amplify viral RNA), we observed a significant inverse correlation (R=–0.696, p=0.0019), implying that higher viral RNA loads (low Ct value) were associated with increased maturation of CMs (*Figure 3F*). We also observed a positive correlation between CD86 expression on nasopharyngeal CMs, and both nasopharyngeal cDC2s (R=0.735, p=0.0012) and cDC1s (R=0.832, p=0.0001) (*Figure 3G and H*, respectively). This indicates that in patients with greater viral RNA load who also have more mature CMs in the nasopharynx, there is a greater likelihood of finding mature MDCs in the nasopharynx as well.

## Elevated cytokine levels observed in plasma and nasopharyngeal secretions correlate with increased monocyte frequencies in the respective compartment

Cytokinemia is a hallmark of severe influenza disease (*Tisoncik et al., 2012*). In order to characterize the degree of inflammation in IAV patients, we measured local and systemic cytokine levels. In agreement with earlier reports (*Cole et al., 2017*; *Oshansky et al., 2014*; *Gill et al., 2008*; *Gill et al., 2005*), we observed elevated levels of TNFα, IL-6, and IFNα in nasopharyngeal secretions (*Figure 4A–C*) as well as in plasma of IAV patients as compared to HCs (*Figure 4D–F*). We also observed elevated levels of plasma IL-10, IL-15, and IL-18 in IAV patients as compared to HCs (data not shown). We compared soluble TNFα levels against frequencies of monocytes and DCs (i.e., potential cellular sources) at the respective anatomical sites; and found a positive correlation between soluble TNFα and CM frequency, both in blood (*Figure 4G*) and in the nasopharynx of IAV patients (*Figure 4H*). However, such correlation was not observed for DCs in blood or NPA (data not shown). Interestingly, we also observed positive associations between age and level of TNFα in plasma and NPA in IAV patients (*Figure 4I*). Furthermore, we observed that plasma IFNα levels were positively correlated with Ct value (suggesting inverse correlation with viral RNA load) (*Figure 4J*). Patients with higher viral RNA loads had lower amounts of IFNα in circulation, which may suggest incomplete protection from infection. For a subset of patients, we assessed chemokines (CCL2, CCL3, and CCL7) in circulation and at the site of infection. Interestingly, in blood, elevated plasma levels of CCL2 correlated positively with IM frequencies (R=0.639, p=0.048), which were significantly elevated in IAV patients, indicating a role for CCL2 in the changes to the IMs during infection (*Figure 4K*). Moreover, during convalescence, TNFα and IL-6 were also reduced both in the nasopharynx and in blood suggesting ablation of both local and systemic inflammation (*Figure 4L*).

## Monocytes are a potent source of systemic TNFα during IAV infection

The localization of mature monocytes and DCs in the nasopharynx during IAV infection and increased cytokine levels in both compartments led us to question whether the cells in circulation were directly involved in inflammation during ongoing infection, or primarily provided a cache of differentiated cells that can migrate to the site of infection. Limited by the number of viable cells obtained from the

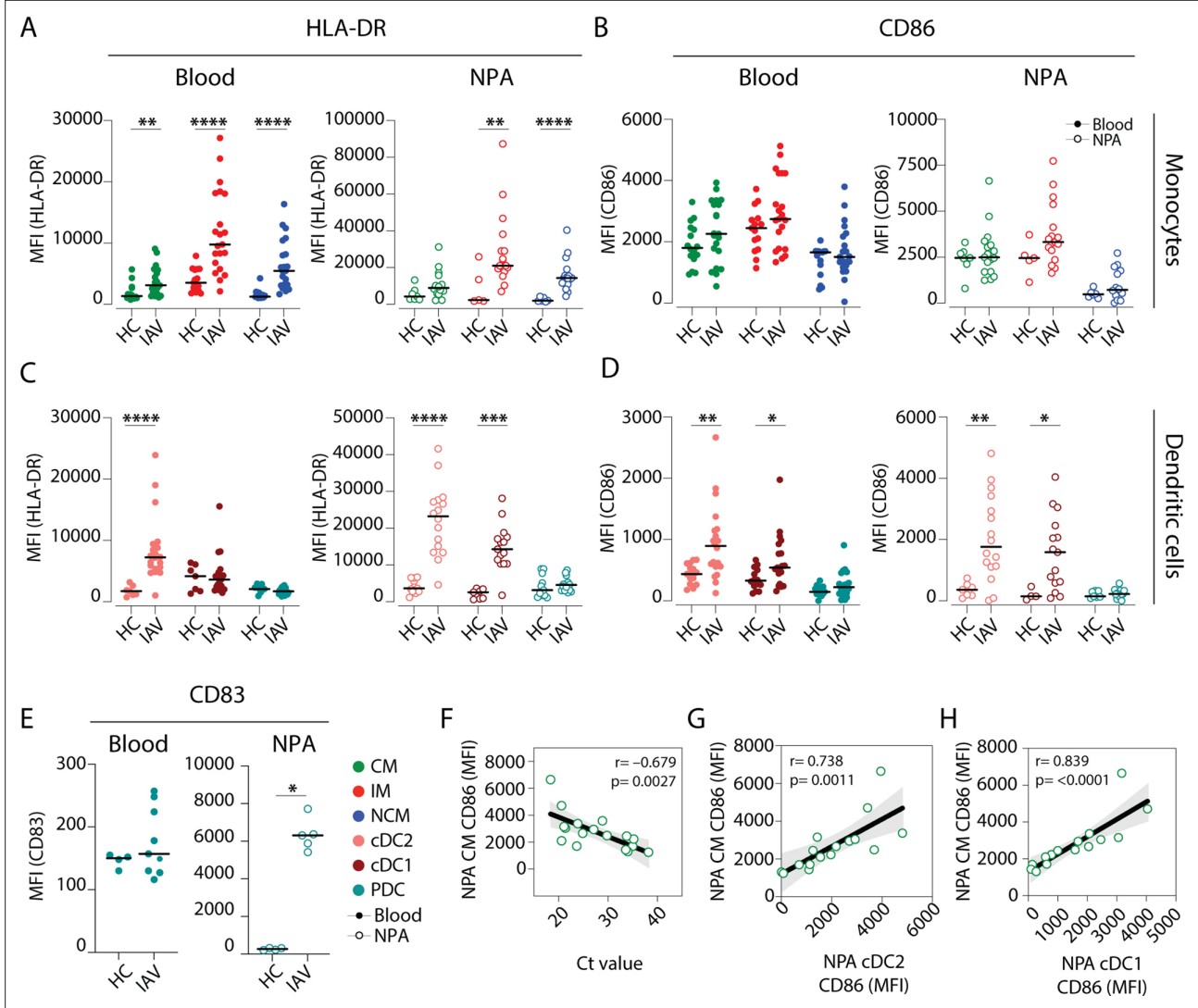

**Figure 3.** Intermediate monocytes and cDCs in the nasopharynx are more mature in influenza A virus (IAV) patients than in healthy controls. (A–D) Scatter plots depict MFI of (A, C) HLA-DR and (B, D) CD86 surface expression on (A–B) monocyte (CD14+CD16– classical monocyte (CM): green, CD14+CD16+ intermediate monocyte (IM); red and CD14–CD16+ nonclassical monocyte (NCM): blue) and (C–D) dendritic cell (DC) (cDC2: coral; cDC1: maroon and plasmacytoid dendritic cell [PDC]: teal) subsets in blood (filled circles) and in the nasopharyngeal aspirates (NPA) (open circles) in healthy controls (HCs) (left, n=11) and IAV patients (right, n=19). (E) Scatter plot depicts MFI of CD83 expression on PDCs in blood (filled circles) and in the NPA (open circles). Differences between IAV patients and HCs were assessed by Mann-Whitney test and considered significant at p<0.05 (*p<0.05, **p<0.01, ***p<0.001, and ****p<0.0001). (F–H) Scatter plots and lines of fit display bivariate linear regression analysis between monocyte maturation status (CD86 surface expression [MFI]) of NPA CMs in IAV+ patients and (F) cycle threshold (Ct) values, (G) maturation status (CD86 surface expression [MFI]) of NPA CD1c+ myeloid DCs (MDCs) in IAV patients and (H) maturation status (CD86 surface expression [MFI]) of NPA CD141+ MDCs in in IAV patients. The shaded area represents the 95% confidence region for the fitted line. R represents Spearman $\rho$ and differences were considered significant at p<0.05.

The online version of this article includes the following source data for figure 3:

**Source data 1.** Related to *Figure 3A*.

**Source data 2.** Related to *Figure 3B*.

**Source data 3.** Related to *Figure 3C*.

**Source data 4.** Related to *Figure 3D*.

**Source data 5.** Related to *Figure 3E*.

**Source data 6.** Related to *Figure 3F*.

**Source data 7.** Related to *Figure 3G*.

**Source data 8.** Related to *Figure 3H*.

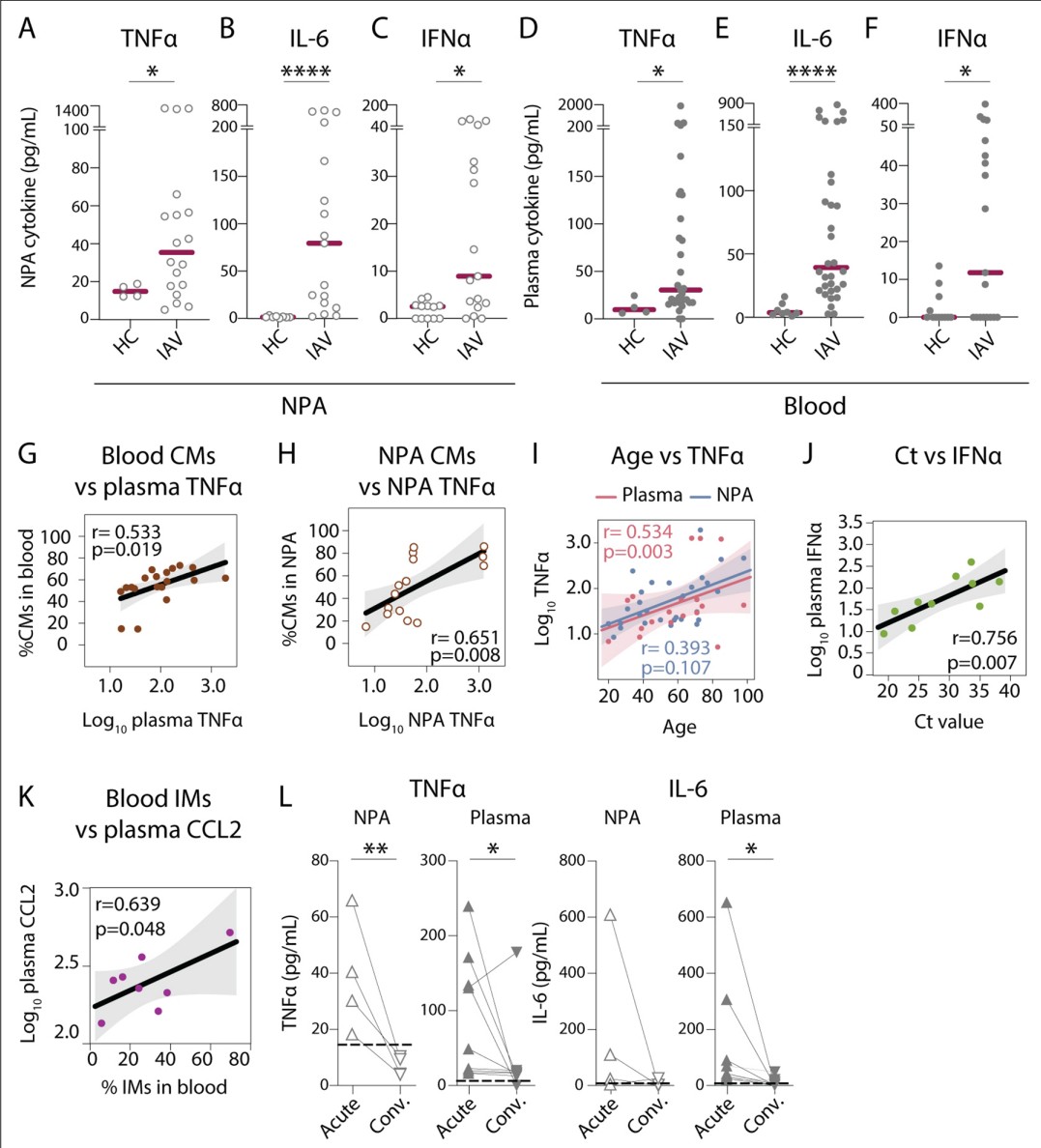

**Figure 4.** Nasopharyngeal and plasma levels of proinflammatory cytokines TNFα, IL-6, and IFNα are elevated during acute influenza A virus (IAV) infection. (**A–F**) Graphs show concentrations of (**A and D**) TNFα, (**B and E**) IL-6, (**C and F**) IFNα in (**A–C**) nasopharyngeal aspirates (NPA) (open circles), and (**D–F**) plasma (filled circles) from healthy controls (HCs) (n=12) and IAV patients (n=31) as determined by ELISA. Lines indicate median concentration. Differences between IAV patients and HCs were assessed using Mann-Whitney test and considered significant at p<0.05 (*p<0.05, **p<0.01, ****p<0.0001). (**G–K**) Scatter plots and lines of fit display bivariate linear regression analysis between variables. The shaded area represents the 95% confidence region for the fitted line. R represents Spearman $\rho$ and differences were considered significant at p<0.05. (**G**) Log$_{10}$ plasma TNFα values vs. CD14+CD16 classical monocyte (CM) frequency in blood of IAV patients; (**H**) log$_{10}$ NPA TNFα vs CM frequency in the NPA of IAV patients; (**I**) age vs. log$_{10}$ TNFα values in plasma (pink) and NPA (blue) in IAV patients; (**J**) cycle threshold (Ct) values vs. log$_{10}$ IFNα values in IAV patients (light green) and (**K**) log$_{10}$ plasma CCL2 vs. intermediate monocyte (IM) frequency in blood of IAV patients. (**L**) Graphs depict TNFα and IL-6 levels in the NPA (n=4) and plasma (n=8) during the acute (upward triangles) and convalescent phase (downward triangles) in IAV patients. Dashed lines depict median frequency values from HCs in blood and NPA. Differences between acute and convalescent phase values were assessed using Wilcoxon matched-pairs signed rank test and considered significant at p<0.05 (*p<0.05 and **p<0.01).

The online version of this article includes the following source data for figure 4:

*Figure 4 continued on next page*

*Figure 4 continued*

**Source data 1.** Related to *Figure 4A*.

**Source data 2.** Related to *Figure 4B*.

**Source data 3.** Related to *Figure 4C*.

**Source data 4.** Related to *Figure 4D*.

**Source data 5.** Related to *Figure 4E*.

**Source data 6.** Related to *Figure 4F*.

**Source data 7.** Related to *Figure 4G*.

**Source data 8.** Related to *Figure 4H*.

**Source data 9.** Related to *Figure 4I*.

**Source data 10.** Related to *Figure 4J*.

**Source data 11.** Related to *Figure 4K*.

**Source data 12.** Related to *Figure 4L*.

nasopharynx, we tested the functional response of circulating monocytes and DCs to in vitro stimulation with a TLR7/8 agonist which mimics ssRNA and then quantified the frequency of TNF-producing cells in each monocyte subset and in cDCs (*Figure 5A and B*). We observed that blood monocytes and DCs from IAV patients produced TNFα spontaneously, in the absence of any external stimulus, with most of the cytokine coming from monocytes (CMs > IMs > NCMs) (*Figure 5C*). In contrast, blood monocytes and DCs from HCs only produced TNFα upon stimulation with TLR7/8L (*Figure 5D*). Importantly, while many of the IAV patients had cells producing TNFα spontaneously, monocytes and DCs remained responsive and had the potential for further increased frequency of TNF-producing cells in response to TLR7/8L stimulation (*Figure 5D*). In summary, our data suggest that during IAV infection, mature monocytes and DCs accumulate in the nasopharynx, and blood monocytes and DCs function as a general source of TNFα, potentially contributing to the systemic inflammatory effects accompanying influenza infections.

## Nasopharyngeal aspirates allow assessment of in situ immune responses to mild infections with other respiratory viruses including SARS-CoV-2

In order to determine whether our findings were specific to IAV or a reflection of immune responses to respiratory viral infections in general, we analysed samples from patients with confirmed IBV, RSV, or SARS-CoV-2 infections. All patient groups were relatively close in age (*Table 1*). SARS-CoV-2 patients tended to seek medical attention later: median 12 days with symptoms (SARS-CoV-2) as compared to 4 days (IAV), 5 days (IBV), and 6 days (RSV) (*Table 1*). Similar to IAV patients, patients with SARS-CoV-2 and RSV had a higher yield of cells in the nasopharynx than HCs (*Figure 6A*). The expansion in monocyte (IM) frequencies in blood we saw in IAV patients was not observed in other infections. Interestingly, in the nasopharynx, monocytes and DCs were significantly elevated during IAV infection, but not during IBV infection (*Figure 6B*). Within the monocyte compartment, we noted differences between the different pathogens. IMs were not increased during IBV or SARS-CoV-2 infection but were increased during RSV infection in both blood and the nasopharynx (*Figure 6C*). The DC compartment too showed differences between the groups in NPA. In blood, all DC subsets were decreased in all groups (*Figure 6D*). cDC2s were increased in the nasopharynx only during IAV and SARS-CoV-2 infections. Interestingly, cDC1s were significantly increased in the nasopharynx only during IAV infection. PDCs in NPA were increased in all groups.

In plasma, individuals with SARS-CoV-2 infection had elevated levels of TNFα in comparison with HCs and IAV patients (*Figure 6E*). SARS-CoV-2 infection was associated with elevated levels of IL-6 compared to HCs, but not compared to the IAV and IBV groups. In NPA, only the IAV and IBV groups had increased levels of IL-6. Despite having increased frequencies of PDCs in NPA, nasopharyngeal levels of IFNα were not elevated in SARS-CoV-2 patients. Together, these data show different patterns of monocyte and DC engagement in the nasopharynx and in blood, and also between IAV, IBV, RSV, and SARS-CoV-2, suggesting a requirement for further scrutiny.

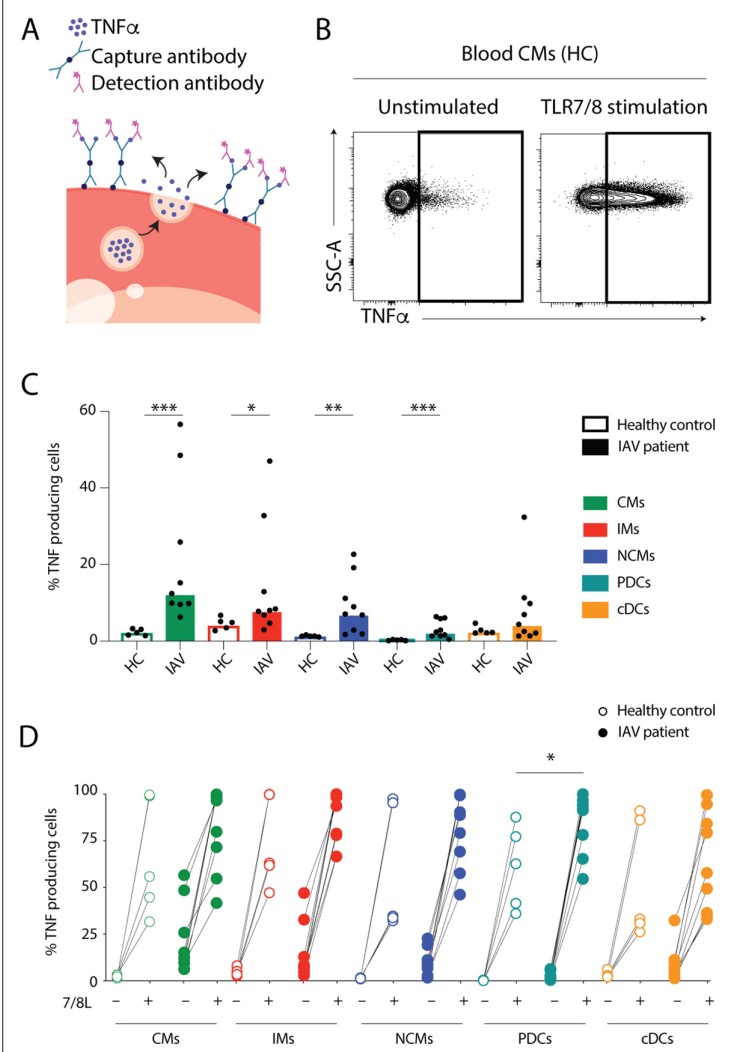

**Figure 5.** Monocytes and dendritic cells (DCs) from influenza A virus (IAV) patients produce TNFα ex vivo without stimulation. (**A**) TNFα release assay: the capture antibody (green) immobilizes secreted TNFα on the cell surface of the TNFα-secreting cell. The PE-labelled detection antibody (pink) together with the phenotypic antibody panel allows detection of TNFα production in individual cell subsets by flow cytometry. (Illustrations were modified from Servier Medical Art, licensed under a Creative Commons Attribution 3.0 Unported License.) (**B**) Representative flow cytometry plots of TNFα-producing blood classical monocytes (CMs) from one healthy control (HC) after 2 hr at 37°C without (unstimulated) or with TLR7/8L stimulation. (**C**) Bar graphs display median frequency of TNFα-producing cells in CD14+CD16– CMs (green), CD14+CD16+ intermediate monocytes (IMs) (red), CD14–CD16+ nonclassical monocytes (NCMs) (blue), plasmacytoid DCs (PDCs) (teal), and total cDCs (orange) in blood in HCs (open, n=5) and IAV patients (filled, n=9) in the absence of TLR stimulation. Each dot represents an individual donor. (**D**) Graph displays frequency of TNFα-producing cells in monocyte and DC subsets from HCs (open, n=5) and IAV patients (filled n=9) in the absence (–) or presence (+) of TLR7/8L stimulation. Differences between IAV patients and HCs in (**C**) were assessed by Mann-Whitney test and in (**D**) with two-way ANOVA using Sidak's multiple comparisons test and considered significant at p<0.05. (*p<0.05, **p<0.01, and ***p<0.001).

The online version of this article includes the following source data for figure 5:

**Source data 1.** Related to *Figure 5C*.

**Source data 2.** Related to *Figure 5D*.

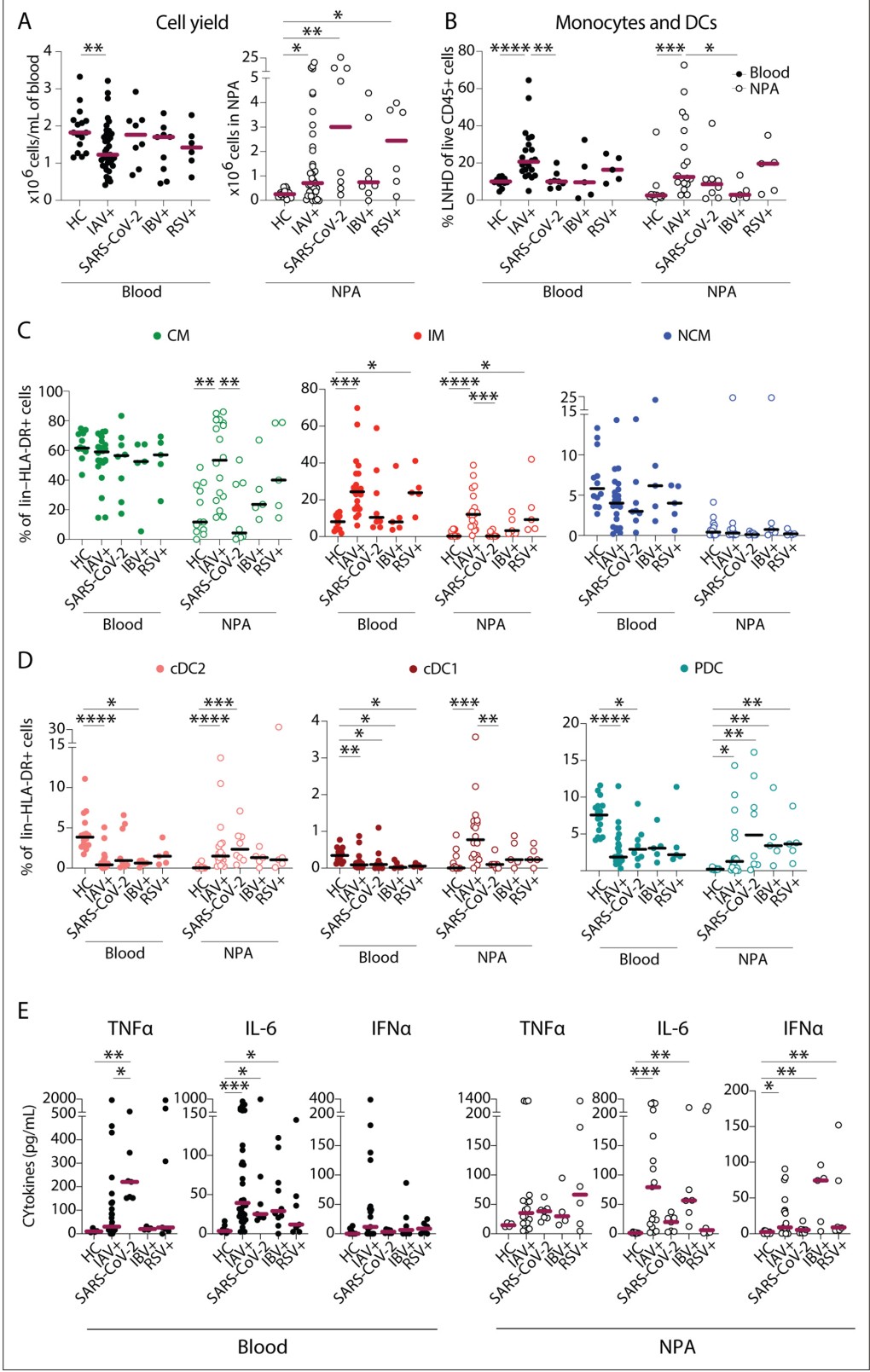

**Figure 6.** Severe acute respiratory syndrome coronavirus 2 (SARS-CoV-2) infection elicits a distinct pattern of innate immune response as compared to influenza A virus (IAV) infection. (**A–B**) Scatter plots show data from individual subjects and lines indicate median values. (**A**) Cell yield per mL blood (×10⁶ PBMCs, filled circles) and total nasopharyngeal aspirates (NPA) cells (×10⁶ cells, open circles) obtained from patients (IAV n=22, SARS-

*Figure 6 continued on next page*

*Figure 6 continued*

CoV-2 n=8, influenza B virus [IBV] n=5 and respiratory syncytial virus [RSV] n=5), and healthy controls [HCs] (n=12). (**B**) Frequency of lineage negative HLA-DR+ cells of live CD45+ cells in blood (filled circles) and NPA (open circles) from HCs and patients. (**C–D**) Scatter plots show frequencies of (**D**) monocyte subsets (CD14+CD16– CM: green, CD14+CD16+ intermediate monocyte [IM]: red, CD14–CD16– nonclassical monocyte [NCM]: blue), and (**E**) dendritic cell [DC] subsets (cDC2: coral, cDC1: maroon, PDC: teal) in PBMCs (filled circles) and NPA (open circles) from HCs and patients. Graphs show data from individual subjects and lines represent median values. Differences between HCs or IAV patients and other patient groups were assessed using Kruskal-Wallis test with Dunn's multiple comparisons test and considered significant at p<0.05 (*p<0.05, **p<0.01, ***p<0.001, and ****p<0.0001). (**E**) Graphs show concentrations of TNFα, IL-6, and IFNα in plasma (filled circles) and NPA (open circles) from HCs and patients as determined by ELISA. Lines indicate median concentration. Differences between IAV patients and HCs were assessed using Mann-Whitney test and considered significant at p<0.05 (*p<0.05, **p<0.01, and ***p<0.001).

The online version of this article includes the following source data for figure 6:

**Source data 1.** Related to *Figure 6A* – blood.

**Source data 2.** Related to *Figure 6A* – nasopharyngeal aspirates (NPA).

**Source data 3.** Related to *Figure 6B* – blood.

**Source data 4.** Related to *Figure 6B* – nasopharyngeal aspirates (NPA).

**Source data 5.** Related to *Figure 6C* – classical monocyte (CM).

**Source data 6.** Related to *Figure 6C* – intermediate monocyte (IM).

**Source data 7.** Related to *Figure 6C* nonclassical monocyte (NCM).

**Source data 8.** Related to *Figure 6D* – cDC2.

**Source data 9.** Related to *Figure 6D* – cDC1.

**Source data 10.** Related to *Figure 6D* – plasmacytoid dendritic cell (PDC).

**Source data 11.** Related to *Figure 6E* – blood.

**Source data 12.** Related to *Figure 6E* – nasopharyngeal aspirates (NPA).

## Innate immunity-related proteins are differentially expressed in plasma and nasopharyngeal secretions of IAV and SARS-CoV-2 infected patients

To further emphasize the distinctive aspects of immune responses between anatomical compartments, and across infections, we utilized SomaScan, an aptamer-based proteomics platform assay to provide a broad assessment of the human proteome (n=7288 analyses targeting 6595 unique human protein measurements) between acute (IAV and SARS-CoV-2) patients and HCs. Based on PCA, plasma and nasopharyngeal proteome composition displayed an overlap regardless of infection or healthy status, separating mainly on the basis of biological site (*Figure 7A*). In IAV patients compared to HCs, more proteins were differentially expressed in blood (n=548) than in the nasopharynx (n=281) (*Figure 7B*), with an overlap of only 8.4% between anatomical sites (n=60 proteins) (*Figure 7C–D*). Interestingly, in SARS-CoV-2 patients compared to HCs, the nasopharynx (n=1868) had considerably more differentially expressed proteins than blood (n=693), over 1500 of which were unique to the nasopharynx. Also, in both anatomical compartments, many differentially expressed proteins were unique to IAV or SARS-CoV-2 (*Figure 7D*, right panels). Pathway analysis was performed to identify significantly enriched GO biological processes and pathways across the four t test models. The number and type of enriched pathways varied across models (*Figure 7E* and *Supplementary file 2*). The largest cluster, primarily mapped to IAV (plasma and NPA), and to a lesser extent, to SARS-CoV-2 NPA, where proteins mapped to host immune responses and innate immunity, supporting the data described elsewhere in this study. Notably, pathways containing TNF, IL-6, ISG15, IL-18R, CCL7, CXCL10 (IP-10), CXCL11, GZMB, SEMA4A, S100A8, S100A9 were all associated with the nasopharynx in IAV patients, with IL-6 pathways showing significant association with nasopharynx in SARS-CoV-2 patients. Both NPA models had the greatest number of significantly enriched pathways (IAV = 51 and SARS-CoV-2=64) compared to the plasma models (IAV = 12 and SARS-CoV-2=19).

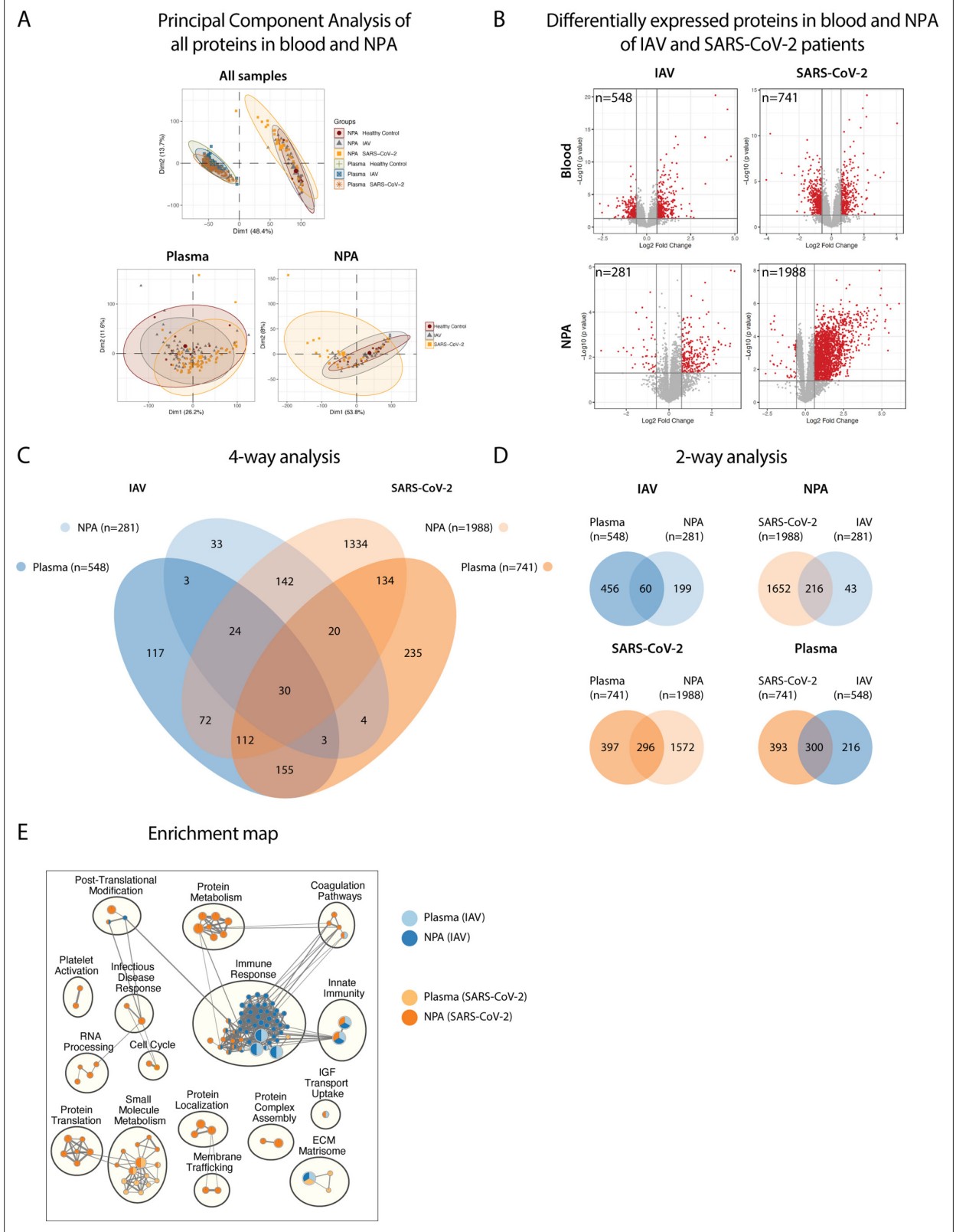

**Figure 7.** Pathway analysis delineates biological pathways and processes in influenza A virus (IAV) and severe acute respiratory syndrome coronavirus 2 (SARS-CoV-2) infection. (**A**) Principal component analysis illustrates differences between proteins in plasma and NPA from 45 IAV and 34 SARS-CoV_2 patients, in comparison with 11 healthy controls, as assayed by the SomaScan proteomic platform. (**B**) Volcano plots show differentially expressed proteins (DEPs) in plasma and nasopharyngeal aspirates (NPA) of both IAV and SARS-CoV-2 patients. N reflects unique EnsembleGene symbols above

*Figure 7 continued*

the threshold (p<0.05 and log$_2$ fold-change >1.5). (**C**) Four-way and (**D**) two-way analysis depicts unique or overlapping DEPs identified in plasma (lighter) and NPA (darker), and between IAV (blue) and SARS-CoV-2 (orange) patients. (**E**) G:profiler pathway enrichment comparing plasma (lighter) and NPA (darker) specimens from patients with IAV (blue) or SARS-CoV-2 (orange) infection. Cytoscape and EnrichmentMap were used for clustering and visualization of the enrichment results. Nodes represent enriched gene sets, clustered with related gene sets according to their gene content. Enrichment results were mapped as a network of gene sets (nodes). Node size is proportional to the total number of genes within each gene set. Proportion of shared genes between gene sets is represented as the thickness of the edge connecting nodes. The network map was manually curated by assigning functional categories to each cluster and by removing singleton gene sets. A complete list of enriched gene sets can be found in Table S2.

## Discussion

In this study, we mapped monocyte and DC distribution and function in blood and in the nasopharynx (the initial site of infection) of patients with ongoing IAV, IBV, RSV, or SARS-CoV-2 infections and in HCs. Several studies have examined immune responses during severe influenza, in particular, focusing on patients sampled during or immediately following the 2009 H1N1 pandemic (*Cole et al., 2017*; *Oshansky et al., 2014*; *Dunning et al., 2018*; *Marion et al., 2016*), or patients hospitalized with severe respiratory symptoms (*Cole et al., 2017*; *Gill et al., 2008*; *Gill et al., 2005*; *Dunning et al., 2018*; *Marion et al., 2016*); and more recently, the immune dysregulation during severe COVID-19 (*Zingaropoli et al., 2021*; *Del Valle et al., 2020*; *Merad and Martin, 2020*; *Mudd et al., 2020*). Here, we elucidated the innate myeloid cell composition and responses in a cohort with relatively mild symptoms, advanced age, and underlying comorbidities, typical of seasonal influenza. Finally, we showed that our methods can be adapted to study immune responses in other viral infections, including the current COVID-19 pandemic.

Flow cytometric characterization revealed an influx of monocytes and DCs to the nasopharynx during infection, in line with previous reports in more severe IAV patients (*Oshansky et al., 2014*; *Gill et al., 2008*; *Gill et al., 2005*; *Herold et al., 2006*). Interestingly, age appeared to differentially skew the CD14+CD16+ IM response in IAV patients. Despite having higher baseline frequencies of circulating CD16-expressing monocytes (*The Milieu Intérieur Consortium et al., 2018*), older patients have fewer CD14+CD16+ IMs in blood during acute IAV infection and concurrently more IMs in the nasopharynx. The increase in IMs suggests a more inflammatory milieu at the site of infection in older patients, perhaps contributing to disease severity as previously suggested in pandemic influenza (*Cole et al., 2017*; *Oshansky et al., 2014*; *Kuiken and Taubenberger, 2008*; *Kuiken et al., 2010*). Despite the general increase in CD14+CD16+ IM frequencies, CD14+CD16– CMs remained the most frequent monocyte/DC subset in either compartment implying a functional role for CMs during IAV infection. Similar to studies in paediatric patients (*Gill et al., 2008*; *Gill et al., 2005*), we also found that over the course of the illness, DCs appeared to migrate from blood to infiltrate the nasopharynx. However, older patients displayed a weaker recruitment of CD1c+ MDCs. In vitro studies have shown that monocytes can differentiate into type I IFN producing mo-DCs in response to IAV exposure (*Cao et al., 2012*). It is therefore possible that CMs recruited to the nasopharynx in older individuals preferentially differentiate into IMs rather than mo-DCs. Moreover, cDC2 function is critical for clearance of IAV infection (*GeurtsvanKessel et al., 2008*). Therefore, diminished recruitment of cDC2s and/or reduced in situ DC differentiation (and therefore delayed or attenuated CD8+ T cell responses) may also contribute more severe disease in older patients.

At the time this study was conceived, DC3s were not yet identified. Guided by a canonical approach and limited by nasopharyngeal cell recovery, we were unable to capture the true heterogeneity of monocyte and DC subsets within the human airways in this study. From later reports establishing DC3 as a distinct DC subset arising from a specific progenitor (*Bourdely et al., 2020*; *Cytlak et al., 2020*; *Dutertre et al., 2019*), especially those describing dysregulated immune responses during SARS-CoV-2 infection (*Winheim et al., 2021*; *Kvedaraite et al., 2021*; *Affandi et al., 2021*), it appears that DC3 may be depleted during severe infection. As the pattern of overall DC depletion from circulation is shared between IAV and SARS-CoV-2 infections, it is plausible that DC3 may be present in the nasopharynx during acute IAV infection. As our comprehension of monocyte and DC heterogeneity expands, it is vital that studies like the present are carried out to accurately map the fate of newly discovered subsets in the airways.

During acute disease, monocytes present in/recruited to the nasopharynx likely contribute to sustained DC recruitment by secreting TNFα, CCL2, CCL3, and CCL7 locally (*Oshansky et al., 2014*;

*Gill et al., 2008*; *Marion et al., 2016*). Due to the limited availability of cells in the NPA samples, demonstrating cytokine secretion from nasopharyngeal cells was not possible. However, we noted that in the nasopharynx, CMs, cDC2s, and cDC2s all appeared to mature by upregulating HLA-DR and/or CD86 expression as compared to cells in HCs, suggesting local activation, likely in response to the inflammatory milieu and/or virus exposure. CD86, a critical costimulatory molecule and ligand for CD28 and CTLA-4, is upregulated on monocytes and DCs upon virus infection in vitro and in vivo (*Hubo et al., 2013*). Lower expression of CD86 on circulating myeloid cells as compared to naso-pharyngeal cells during IAV infection suggests migration of mature cells to the nasopharynx, and/or in situ differentiation of monocytes to induced DCs (that may not upregulate maturation markers) in blood during acute IAV infection as previously speculated (*Cao et al., 2012*). A non-human primate model of chronic simian immunodeficiency virus infection revealed sustained mobilization of MDCs from the bone marrow via blood to the intestinal mucosa, where MDCs remained activated and eventually underwent apoptosis (*Wijewardana et al., 2013*). In IAV infection, monocytes and DCs are likely recruited to the nasopharynx for a shorter period of time, where they face a similar fate. Some DCs likely traffic antigens to the lymph node to support adaptive responses (*Cella et al., 1999*; *GeurtsvanKessel et al., 2008*).

Nasopharyngeal CMs were also more mature in patients with higher viral RNA loads and their recruitment and/or differentiation may be aided by local cytokine production (TNFα and IL-6). Key innate inflammatory cytokines, including TNFα, IL-6, IFNα, IL-10, IL-15, and IL-18, were all significantly elevated early during infection (days 1–5) compared to HCs. TNFα levels strongly correlated with the presence of CMs in both blood and nasopharynx; despite the significant expansion of IMs in blood, supporting previous studies (*Cole et al., 2017*; *Boyette et al., 2017*; *Dunning et al., 2018*; *Sprenger et al., 1996*). We also demonstrated that the circulating CMs were the most frequent cellular source of TNFα among blood monocytes and DCs, while the TNFα at the site of infection may come from both CMs and the IMs (both subsets found in increased frequencies). Ex vivo, at steady state, human blood CMs and IMs have comparable TNFα secretion irrespective of the nature of stimulation (*Boyette et al., 2017*). IAV infection, therefore, may skew monocyte subset function, in addition to monocyte distribution and maturation. The broad range of TNFα, IL-6, and IFNα responses seen among the IAV patients suggest that seasonal IAV strains are associated with milder cytokine responses than those observed during infections with 2009 H1N1 pandemic influenza or highly pathogenic avian H5N1 strains (*Oshansky et al., 2014*; *Sakabe et al., 2011*; *Dunning et al., 2018*; *Gerlach et al., 2013*). Within the setting of seasonal influenza however, older patients had more IMs in the NPA and displayed stronger TNFα responses, locally and systemically.

During IAV infection, the significant expansion of CD14+CD16+ IMs in blood and the nasopharynx could be mediated by the presence of CCL2, a chemokine critical for monocyte chemotaxis (*Dean et al., 2008*). Upon CCL2 ligation, CCR2 facilitates CM and IM adhesion and transmigration from blood to tissues (*Tsou et al., 2007*). Therefore, in IAV patients, CCL2 may contribute to recruitment of monocytes and DCs from circulation to the nasopharynx to aid inflammation (*Oshansky et al., 2014*; *Sprenger et al., 1996*). Furthermore, elevated IFNα in circulation correlated with reduced viral RNA load, although we saw no evidence of this locally at the site of infection, suggesting that perhaps lower systemic IFN responses can indicate impaired control of viral replication in the nasopharynx, as previously reported (*Dunning et al., 2018*). This finding also supports previous studies showing that attenuated RIG-I signalling impairs IFN responses in the elderly (*Molony et al., 2017*) and very young (*Marr et al., 2014*), both groups at high risk of influenza-associated mortality.

During convalescence, monocyte and DC frequencies, and cytokine levels normalized in blood and the nasopharynx to levels seen in healthy individuals. The local and systemic inflammation observed, therefore, appears limited to the acute phase of disease and wanes during convalescence. Unlike patients infected with the pandemic 2009 H1N1 IAV strain during the pandemic years (2009–2010), where long-lasting immune perturbations were documented *Lichtner et al., 2011*; the repercussions of relatively mild seasonal IAV infections appear short-lived. This is also in contrast to in vivo studies which demonstrated prolonged DC activation and disruption of DC subset reconstitution in the lungs following IAV infection in mice (*Strickland et al., 2014*).

In April-May 2020, during the COVID-19 pandemic, the approach and methods described here allowed us to implement this study plan in a rapidly evolving pandemic. Although the patient cohorts were not identical, this endeavour proved the feasibility of using nasopharyngeal aspiration to assess

local immune responses during respiratory viral infections. We observed that patients with IBV, RSV, or SARS-CoV-2 also displayed an influx of cells to the nasopharynx during infection. Despite annual reoccurrence of IBV and RSV infections, detailed investigations of the innate cells at the site of infection in patients are quite rare (*Gill et al., 2008*; *Gill et al., 2005*), and the technique we describe allows for minimally invasive longitudinal measurement of cellular and molecular aspects of infection. Patients with mild SARS-CoV-2 infection have not been studied as extensively as patients with severe or fatal COVID-19. The cell influx to the nasopharynx in COVID-19 patients did not appear to be due to monocytes but rather due to cDC2s and PDCs, which were reduced in blood, in line with previous reports (*Zingaropoli et al., 2021*; *Peruzzi et al., 2020*). Also in line with previous studies (*Del Valle et al., 2020*; *Merad and Martin, 2020*; *Mudd et al., 2020*), we found elevated levels of IL-6 in plasma, and also of TNFα. As different SARS-CoV-2 variants emerge, with concurrently increasing global vaccination rates, incomplete neutralization and/or protection and herd immunity may result in recurring mild SARS-CoV-2 infections. Understanding site-of-infection responses to mild SARS-CoV-2 infections is therefore increasingly relevant, and can be accomplished by the methods we describe in our study.

Finally, to further highlight the importance of studies like ours, we employed a high sensitivity, high-throughput proteomics platform to quantify differences in the nasopharyngeal proteome as compared to blood, in both IAV and SARS-CoV-2 infections. Additionally, this approach allows identification of proteins that cannot be identified by conventional assays that may be limited in breadth of proteome coverage or by limits of detection. We found differentially expressed proteins unique to blood or nasopharynx, and also to IAV or SARS-CoV-2. Preliminary pathway analysis reveals sizeable changes within the nasopharyngeal proteome during infection, especially during SARS-CoV-2 infection. In IAV patients, in line with previous reports (*Marion et al., 2016*) several innate immune pathway proteins were differentially enriched in the nasopharyngeal proteome, including TNF and IL-6, as seen from ELISA data.

The present study augments our current understanding of the role of monocyte and DC subsets during human respiratory viral infections by highlighting unique dynamics and location-specific functions for each subset over the course of infection. Acute human IAV infection is characterized by an expansion of IMs and monocyte-mediated cytokinemia driving monocyte and DC migration from blood to the nasopharynx, the initial site of IAV infection. Older patients with comorbidities display a recruitment pattern skewed towards IMs and increased TNF rather than CD1c+ MDCs which may contribute to more severe disease and longer duration of hospitalization observed in these patients. IBV, RSV, and mild SARS-CoV-2 infections elicit different patterns on immune cell recruitment to the nasopharynx as compared to IAV infections. We also illustrate the value of comparative high-resolution studies of immune cells in blood and at the site of infection, in order to fully understand their individual contributions to disease and how they orchestrate inflammation synergistically. Similar studies, carried out longitudinally across tissues, will aid resolution of these findings, and allow multivariate modeling of biomarkers of disease severity. Therapeutic approaches which allow selective modulation of monocyte and DC redistribution, maturation, and cytokine/chemokine function may hold the key to reducing influenza-associated disease burden and mortality in the future.

## Methods
### Study subjects
Patients seeking medical attention for ILI at the Emergency Department at the Karolinska University Hospital in Stockholm, Sweden, during three consecutive influenza seasons (January-March) of 2016–2018 were recruited to the study following informed consent. The inclusion criteria for enrolment of patients were (1) age >18 years, (2) no known immunodeficiency, (3) not taking antibiotics, immunomodulatory, or anti-inflammatory medication at time of inclusion, (4) presenting with fever and at least one of the following symptoms of ILI: cough, nasal congestion, headache, or muscle ache. For the current study, only data from patients who had confirmed IAV, IBV, RSV, or (later) SARS-CoV-2 infections were used. Convalescent samples were collected after at least 4 weeks, ensuring absence of respiratory symptoms in the prior week. Healthy volunteers were recruited and sampled similarly outside of influenza season. During the COVID-19 pandemic, additional patients were included between April and May 2020 at the Infectious Diseases ward at the Karolinska University Hospital or the Haga Outpatient Clinic (Haga Närakut) in Stockholm, Sweden, as well as mild/asymptomatic

household contacts of patients with confirmed COVID-19 were screened by PCR and enrolled if positive. Clinical data were obtained from the patients and medical records (*Table 1*) and are extensively discussed in a previous publication (*Falck-Jones et al., 2021*). Total burden of comorbidities was assessed using the CCI (*Charlson et al., 1987*).

The severity of disease was categorized using the respiratory domain of the sequential organ failure assessment (SOFA) score (*Vincent et al., 1996*). In the absence of arterial partial pressure of oxygen (PaO$_2$), peripheral transcutaneous haemoglobin saturation (SpO$_2$) was used instead to calculate a modified SOFA score (mSOFA) (*Grissom et al., 2010*). Fraction of inspired oxygen (FiO$_2$) estimation based on O$_2$ flow was done in accordance with the Swedish Intensive Care register definition (*Intensivvårdsregistret, 2018*). Mild disease was defined as PaO$_2$/FiO$_2$ (PFI)>53 kPa (>400 mmHg) or SpO$_2$/FiO$_2$ (SFI)>400. Moderate disease was defined as PFI >27–53 kPa (>200–400 mmHg) or SFI 235–400. The disease severity score is more extensively described in a previous publication (*Falck-Jones et al., 2021*).

## Sample collection

Blood, nasal swabs, and nasopharyngeal aspirates (NPA) were obtained from all patients (acute and convalescent phase samples) and HCs (*Figure 1A*). Briefly, up to 30 mL venous blood was collected in Vacutainer tubes containing EDTA, for blood counts and PBMC isolation. Nasopharyngeal swabs (Sigma Virocult) were collected for diagnostic qPCR. NPA samples were collected into a vacuum trap by inserting a thin catheter through the naris, deep into the nasopharynx and applying gentle suction for 1–3 min. The vacuum trap and tubing were rinsed out with 3 mL sterile PBS. All samples were processed within 2 hr of sampling.

## Diagnostic tests to determine etiology of infection

Nasal swab samples were analysed for IAV, IBV, and RSV by real-time PCR using the commercial Simplexa system (*Svensson et al., 2014*), as well as bacteria (by culture methods). The tests were performed at the Department of Clinical Microbiology, Karolinska University Laboratory as part of routine diagnostics for respiratory viral infections. Ct values from the Simplexa assay were considered as semi-quantitative measures of virus levels in statistical analyses. Bacterial culture results were retrieved wherever available from the Department of Clinical Microbiology at the Karolinska University Hospital. Convalescent individuals and HCs were confirmed IAV– by qPCR. SARS-CoV-2 infection was diagnosed similarly using the GeneXpert SARS-CoV-2 detection system (Cepheid). Supplementary data on bacterial cultures were retrieved from the microbiology lab/clinical records.

## Isolation of cells from blood and nasopharyngeal aspirates

Blood samples were centrifuged at 800 g/10 min/room temperature (RT) and plasma was frozen at –20°C. The blood volume was reconstituted with sterile PBS and PBMCs were obtained by density-gradient centrifugation using Ficoll-Paque Plus (GE Healthcare) after centrifugation at 900 g/25 min/RT (without brake). NPA samples were centrifuged at 400 g/5 min/RT and the supernatant was frozen at –20°C. The cells were washed with 5 mL sterile PBS to remove mucus, by filtering through a 70 µm cell strainer followed by centrifugation for 400 g/5 min/RT. Cell counts and viability were assessed using Trypan Blue (Sigma) exclusion and an automated Countess cell counter (Invitrogen) or manual counting using a light microscope (when the counter was unable to distinguish live and dead NPA cells). Cells were used fresh for flow cytometry (all NPA cells when samples contained >10$^5$ cells and had ≥70% viability; and at least 1×10$^6$ PBMCs) or frozen in RNALater (<10$^5$ NPA cells) (Thermo Fisher). Excess PBMCs were cryopreserved in FBS (Gibco) + 10% DMSO (Sigma) and stored in liquid nitrogen.

## Flow cytometry analysis

Phenotypic analysis was performed on NPA cells and PBMCs using Live/Dead Blue; lineage markers CD3 (SK7; BD), CD19 (HIB19; Biolegend), CD20 (L27, BD), CD45 (HI30; BD), CD56 (HCD56, BD) and CD66abce (TET2, Miltenyi Biotec); HLA-DR (TU36, Life Technologies), CD14 (M5E2, BD), CD16 (3GE, Biolegend), CD11c (B-Ly6, BD), CD1c (AD5-8E7; Miltenyi), CD141 (AD5-14H12; Miltenyi), CD123 (7G3; BD); maturation markers CD83 (HB15e, Biolegend) and CD86 (2331; BD); adhesion marker CD62L (SK11, BD); and migration markers CCR2 (K036C2, Biolegend) and CCR7 (150503: BD); and

fixed with 1% paraformaldehyde for flow cytometry. Samples were acquired on an LSRFortessa flow cytometer (BD Biosciences) and analysed using FlowJo software v10 (Tree Star).

## TNF-α release assay

TNFα secretion from fresh PBMCs in response to TLR stimulation was assessed using the TNF-α Secretion Assay-Detection Kit (PE) (130-091-268, Miltenyi Biotec) according to the manufacturer's instructions. TNFα secretion was measured over 2 hr of incubation at 37°C with shaking (200 rpm) in the presence or absence of 1 µg/mL 3M-019 (7/8L) (Invivogen). Briefly, the capture antibody immobilized all secreted TNFα on the surface of the cell, and the detection antibody (PE-labelled) was incorporated into the above panel of antibodies to determine the cellular source of TNFα. Secretion of TNFα was quantified using flow cytometry.

## ELISA and Luminex

Plasma samples and NPA supernatants were assayed for soluble markers using ELISA and Luminex. Human IFNα All Subtype ELISA was performed according to the manufacturer's instructions (PBL Assay Science). TNFα and IL-6 ELISAs were performed using DuoSet kits (R&D Systems). Luminex assays were performed using custom-designed 9-plex (IFNγ, IL-8, IL-10, IL-18, CCL2, CCL3, CCL7, IL-1β, and IL-12p70) kit (R&D Systems) and analysed on the Bio-Plex 200 instrument (Bio-Rad).

## SomaScan assay

Proteomic profiling of NPA and EDTA plasma samples was conducted using SomaScan v4.1 (SomaLogic). The SomaScan proteomic platform has been described previously (*Rohloff et al., 2014*). Briefly, the platform utilizes DNA-based binding reagents (slow off-rate modified aptamers) to quantify by fluorescence the availability of binding epitopes (which represents protein abundance, shape, and charge) for over 7000 protein targets with high specificity and limits of detection comparable to antibody-based assays. The assay captures both high- and low-abundance proteins over a dynamic range of detection of approximately 10 logs (*Williams et al., 2022*). Normalization of SomaScan data was performed on plasma and NPA samples separately for all statistical analysis except PCA visualization. This was done because of the dramatically different signal distributions observed for the two anatomical compartments. Median scaling of samples to the global population median was performed. Additional normalization methods were evaluated and resulted in similar outcomes and performance (data not shown). Only samples derived from patients with a disease severity ≤ 4 (moderate severity) were evaluated. Outlier samples were detected and removed based on correlation with other samples. All pairwise correlations for a given sample with other samples in the same tissue type were calculated. If the median pairwise correlation for a given sample fell below 3 robust standard deviations (MAD) of the median pairwise correlation for all samples that sample was considered an outlier and removed from t test and pathway analysis (see below).

## Statistics

Data were analysed using GraphPad Prism v6.0 (GraphPad Software), JMP, v14.2 (SAS Institute Inc, Cary, NC, 1989–2019) and R (v4.0.3). Differences between frequencies in IAV patients and HCs were assessed using nonparametric tests after assessing normality, using the Mann-Whitney test (at 95% confidence intervals). For comparisons between exposure conditions, two-way ANOVA with Sidak's multiple comparisons test was applied. For comparisons between acute and convalescent phase data, Wilcoxon matched-pairs signed rank test was used. Bivariate and multivariate linear regression analysis was performed using JMP, choosing Spearman's rank correlation coefficient for nonparametric analyses. Differences between HCs or IAV patients and other patient groups were assessed using nonparametric tests after assessing normality, using the Kruskal-Wallis test with Dunn's multiple comparisons test (at 95% confidence intervals). Data were considered significant at $p < 0.05$. Principal components analysis was performed to visualize the global impact of NPA and plasma tissue type and the impact of either IAV or SARS-CoV-2 infection. Non-parametric ellipses cover 95% of samples within a group; t tests were performed comparing the acute phase samples with HCs. Test populations were stratified by tissue type and infectious agent. Differentially expressed proteins were selected based on a $p < 0.05$ and absolute fold-change >1.5 criteria. In cases where individual SOMAmers may interrogate more than one gene product (EnsembleGene) the different genes were separated for

Venn diagrams and pathway analysis. Final gene lists are made up of unique gene lists because of the one-to-many and many-to-one relationship between SOMAmer reagents and protein/gene products.

### Pathway analysis

Analysis of enriched pathways was performed for the four t test comparisons representing IAV and SARS-CoV-2 from either NPA or plasma specimens. Differentially expressed proteins detected by t test (p<0.05) and absolute fold-change >1.5 were used as input for the g:profiler pathway enrichment method using gprofiler2 v0.2.1 (*Kolberg et al., 2020*; *Raudvere et al., 2019*) in R. Annotated gene sets representing Gene Ontology biological processes, Hallmark pathways, C6 Oncogenic signature gene sets, and Canonical Pathways (incl. Biocarta, KEGG, PID, Reactome, and Wikipathways) were downloaded from the Molecular Signatures Database v7.5.1 for use as target pathways (*Liberzon et al., 2011*). Due to the semi-targeted nature of the SomaScan protein quantification method, a unique list of all proteins (ensembleGeneSymbol) interrogated on the platform (n=6370) were used as a statistical null background. An FDR threshold of q<0.05 was used to select enriched gene sets for further analysis. The Cytoscape Enrichment Map plugin v3.3.4 (*Merico et al., 2010*) was used for clustering and visualization of significantly enriched gene sets detected for each cell line xenograft model. Default clustering parameters were applied. Singleton gene sets were removed from visualization if only a single t test comparison contributed to enrichment and the gene set was not connected to a larger functional cluster.

### Study approval

Informed consent was obtained from all patients and volunteers following verbal and written information. The study was approved by the Swedish Ethical Review Authority (No. 2015/1949-31/4) and performed according to the Declaration of Helsinki.

## Acknowledgements

We thank the patients and healthy volunteers who have contributed clinical material to this study. We would like to thank Adeline Mawa, Kevin Fallahi, Marcus Nordin, Roosa Vaitiniemi, Eric Åhlberg, and the research nurses at the Emergency and Infectious Diseases Departments for technical assistance. This work was supported by grants to AS-S from the Swedish Research Council (VR), the Swedish Heart-Lung Foundation, the Bill & Melinda Gates Foundation, the Swedish Childhood Cancer Fund, and Karolinska Institutet.

## Additional information

### Competing interests

Clare Paterson: is an employee and stockholder of SomaLogic Inc. Penn Whitley: is a consultant for Boulder Bioconsulting, Inc. Anna Smed-Sörensen: is a consultant to Astra-Zeneca on studies not related to the present study. The other authors declare that no competing interests exist.

### Funding

| Funder | Grant reference number | Author |
|---|---|---|
| Vetenskapsrådet | | Anna Smed-Sörensen |
| Hjärt-Lungfonden | | Anna Smed-Sörensen |
| Bill & Melinda Gates Foundation | | Anna Smed-Sörensen |
| Barncancerfonden | | Anna Smed-Sörensen |
| Karolinska Institutet | | Anna Smed-Sörensen |

The funders had no role in study design, data collection and interpretation, or the decision to submit the work for publication.

## Author contributions
Sindhu Vangeti, Conceptualization, Data curation, Formal analysis, Investigation, Visualization, Methodology, Writing – original draft, Project administration, Writing – review and editing; Sara Falck-Jones, Meng Yu, Data curation, Formal analysis, Investigation, Methodology, Writing – review and editing; Björn Österberg, Data curation, Investigation, Writing – review and editing; Sang Liu, Data curation, Investigation, Methodology, Writing – review and editing; Muhammad Asghar, Klara Sondén, Resources, Data curation, Supervision, Validation, Investigation, Writing – review and editing; Clare Paterson, Penn Whitley, Data curation, Software, Formal analysis, Writing – review and editing; Jan Albert, Resources, Data curation, Formal analysis, Supervision, Validation, Investigation, Methodology, Writing – review and editing; Niclas Johansson, Conceptualization, Resources, Data curation, Formal analysis, Supervision, Validation, Investigation, Methodology, Project administration, Writing – review and editing; Anna Färnert, Conceptualization, Resources, Data curation, Validation, Investigation, Visualization, Methodology, Project administration, Writing – review and editing; Anna Smed-Sörensen, Conceptualization, Resources, Data curation, Formal analysis, Supervision, Funding acquisition, Validation, Investigation, Visualization, Methodology, Writing – original draft, Project administration, Writing – review and editing

## Author ORCIDs
Sindhu Vangeti (iD) http://orcid.org/0000-0003-3404-6878
Jan Albert (iD) http://orcid.org/0000-0001-9020-0521
Anna Smed-Sörensen (iD) http://orcid.org/0000-0001-6966-7039

## Ethics
Human subjects: Informed consent was obtained from all patients and volunteers following verbal and written information. The study was approved by the Swedish Ethical Review Authority (No. 2015/1949-31/4) and performed according to the Declaration of Helsinki.

## Decision letter and Author response
Decision letter https://doi.org/10.7554/eLife.77345.sa1
Author response https://doi.org/10.7554/eLife.77345.sa2

---

# Additional files

## Supplementary files
- Supplementary file 1. Flow cytometry panels.
- Supplementary file 2. List of enriched pathways from gprofiler.
- Transparent reporting form

## Data availability
Source Data files have been provided for figures 1-7 and Appendix 2_Table 2 as separate excel files.

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

# Appendix 1

**Appendix 1—table 1.** Flow cytometry panels.

**Panel 1: Phenotyping of MNPs from PBMCs**

| Fluorochrome | Marker | Company | Clone |
| --- | --- | --- | --- |
| FITC | CD83 | BioLegend | HB15e |
| PerCp Cy5.5 | CD123 | BD | 7G3 |
| PE Cy7 | CD1c | Miltenyi | AD5-8E7 |
| PE Cy5 | CD11c | BD | B-Ly6 |
| PE TR | HLA-DR | Life Technologies | TU36 |
| PE | CD141 | Miltenyi | AD5-14H12 |
| | CD3 | BD | SK7 |
| | CD19 | BioLegend | HIB19 |
| | CD20 | BD | L27 |
| | CD56 | BD | HCD56 |
| APC-Cy7 | CD66abce | Miltenyi | TET2 |
| AF700 | CD16 | BioLegend | 3GE |
| APC | CD45 | BD | HI30 |
| BV786 | CD62L | BD | SK11 |
| BV650 | CD86 | BD | 2331 |
| BV605 | CCR2 | BioLegend | K036C2 |
| BV510 | CD14 | BD | M5E2 |
| V450 | CCR7 | BD | 150503 |
| DAPI | Live/Dead Blue | Thermo Fisher | Cat no: L34962 |

**Panel 2: Phenotyping of PBMCs following TLR stimulation-TNF-release assay**

| Fluorochrome | Marker | Company | Clone |
| --- | --- | --- | --- |
| FITC | CD83 | Biolegend | HB15e |
| PerCp Cy5.5 | CD123 | BD | 7G3 |
| PE Cy7 | CD1c | Miltenyi | AD5-8E7 |
| PE Cy5 | CD11c | BD | B-Ly6 |
| PE TR | HLA-DR | Life Technologies | TU36 |
| PE | **TNF-α detection antibody** | Miltenyi | (Reagent from 130-091-268) |
| | CD3 | BD | SK7 |
| | CD19 | BioLegend | HIB19 |
| | CD20 | BD | L27 |
| | CD56 | BD | HCD56 |
| APC-Cy7 | CD66abce | Miltenyi | TET2 |
| AF700 | CD16 | BioLegend | 3GE |
| APC | CD141 | Miltenyi | AD5-14H12 |
| BV786 | CD62L | BD | SK11 |
| BV650 | CD86 | BD | 2331 |
| BV605 | CCR2 | BioLegend | K036C2 |
| BV510 | CD14 | BD | M5E2 |

*Appendix 1—table 1 Continued on next page*

*Appendix 1—table 1 Continued*

**Panel 1: Phenotyping of MNPs from PBMCs**

| Fluorochrome | Marker | Company | Clone |
|---|---|---|---|
| V450 | CCR7 | BD | 150503 |
| DAPI | Live/Dead Blue | Thermo Fisher | Cat no: L34962 |

