## [Editor Report]

This study presents a valuable evaluation of the distribution of monocytes and dendritic cells in the blood and nasopharyngeal aspirates of patients with mild respiratory tract infections. They report solid differences between monocytes and dendritic cells and variation with patient age, showing that during influenza A virus infection, the classical and intermediate monocyte numbers increased both, in blood and nasopharyngeal aspirates while DCs increased in the nasopharyngeal aspirates only. The work will be of broad interest to immunologists, lung biologists and infection disease community.

---

## [Decision Letter]

**Decision letter after peer review:**

Thank you for submitting your article "Human influenza virus infection elicits distinct patterns of monocyte and dendritic cell mobilization in blood and the nasopharynx" for consideration by *eLife*. Your article has been reviewed by 2 peer reviewers, and the evaluation has been overseen by a Reviewing Editor and Satyajit Rath as the Senior Editor. The following individual involved in the review of your submission has agreed to reveal their identity: Matthew Collin (Reviewer #2).

Essential revisions:

1) Data are not at the current state of knowledge (naming, gating, subsets). The authors should update their analysis and gating strategy. In particular, the new subset DC3 should be investigated.

2) Supplement with new data to align with contemporary classification, including RNA-seq.

*Reviewer #2 (Recommendations for the authors):*

The manuscript was very well written and interesting to read although the data are highly descriptive. The main limitation was that the classification of myeloid cells is simply outdated. I accept that CD34+ cells would be excluded by CD11c but this gate really does exclude some DC1 which have low CD11c expression. The intermediate monocyte is really just an activated classical monocyte. This caveat could be discussed and perhaps some supporting data put in to show an alternative way of analysing the classical monocyte pool, using CD16 as a marker of activation, rather than trying to define a subset from the smear. SLAN is a good marker for the 'true' non-classical population but this would mean new experiments. DC2 and 3 can be separated by CD14 and CD1c as Jahnsen and colleagues previously showed with their description of CD1c+ 'monocytes' (DC3) 10.1371/journal.pone.0157387. A lot has been written on these recently but it is tricky to split properly without CD163, CD5 or BTLA. The CD123+ population can be displayed against CD11c to identify the space where AX^L+^SIGLEC6+ pre-DC sit (CD11c slightly positive) but again, easier with a more specific pDC antigen or AXL.

---

## [Author Response]

Essential revisions:1) Data are not at the current state of knowledge (naming, gating, subsets). The authors should update their analysis and gating strategy. In particular, the new subset DC3 should be investigated.

The flow cytometry data in this study was obtained already 2016-2018 and the FACS panel was not developed to study the more recently described DC3 subset. Therefore, the panel lacks markers used to identify the DC3 subset, such as CD163, BTLA and CD5 (Cytlak et al., https://doi.org/10.1016/j.immuni.2020.07.003, Villani et al., https://doi.org/10.1126/science.aah4573). In an effort to probe further into existing data despite the shortcomings of the panel, we enumerated CD14+CD1c+ “mo-DC” in the revised version of the manuscript, this is further discussed below in the response to Reviewer 1, and we have added additional text in the manuscript discussing the potential role of DC3s in IAV infection (details below, within responses to reviewer comments).

2) Supplement with new data to align with contemporary classification, including RNA-seq.

We agree with both Reviewers and the Editors that the manuscript would indeed benefit from the inclusion of RNAseq data. However, the generation of RNAseq data is unfortunately beyond the scope of the current manuscript for several reasons. Bulk RNAseq analyses require high-quality RNA isolated from cells of interest. While this is now optimized to do with PBMCs, in contrast, NPA samples have fewer cells, lower viability and a lower proportion of immune cells (our data as well as Deprez et al., https://doi.org/10.1164/rccm.201911-2199OC). In the current study, NPA cells were used for flow cytometry analysis, leaving no cells for other analyses such as RNAseq. Inclusion of new patients in the current study and collection of additional samples is not feasible. Additionally, there is a high likelihood of prior exposure(s) to SARS-CoV-2 in influenza patients sampled in the future. In a preprint, Cheong et al. (https://doi.org/10.1101/2022.02.09.479588) recently showed that patients with severe COVID have persistently high levels of neutrophil progenitors and lasting changes in monocyte phenotypes late into convalescence (4-12 months). In the same study, lasting epigenetic activation signature of inflammatory programs, neutrophil differentiation and monocyte phenotypes were reported in CD14+ monocytes. The authors proposed a durable program of heightened IRF/STAT-driven anti-viral response in monocytes (and matching hematopoietic stem cells) with implications for subsequent infections after recent SARS-CoV-2 infection.

Despite our inability to perform RNAseq analysis due to reasons described above, we were able to use matched NPA and plasma samples from influenza and COVID-19 patients as well as pre-pandemic healthy controls for additional proteomics assays, using the SomaScan platform (new Figure 7 and associated text). This data does not contribute to the classification of different monocyte and DC subsets but instead highlights the importance of looking in the right anatomical location to better understand immune responses to respiratory viral infections. In fact, the central theme in this manuscript is the importance of studying local immune responses at the site of infection during viral respiratory infections and to not restrict analyses to blood. We believe our message is strengthened by the striking differences in proteomic profiles we report in different anatomical compartments.

Reviewer #2 (Recommendations for the authors):The manuscript was very well written and interesting to read although the data are highly descriptive. The main limitation was that the classification of myeloid cells is simply outdated. I accept that CD34+ cells would be excluded by CD11c but this gate really does exclude some DC1 which have low CD11c expression.

As previously mentioned in our response to Reviewer 1, an important feature of this study was our efforts to describe monocytes and dendritic cells (DC) in the human nasopharynx during influenza A virus infection, and provide a comparison with healthy and convalescent individuals. Further, we wished to emphasize the value of studying the nasopharynx during respiratory viral infections, particularly in light of the ongoing COVID-19 pandemic. We described a non-invasive method to (longitudinally) sample this anatomical compartment, that allows retrieval of intact immune cells as well as mucosal fluid for soluble marker analysis. As Reviewer 2 also pointed out, there are some drawbacks in the current study.

Primarily, a majority of the flow cytometry data was generated on samples collected during the influenza seasons of 2016-2018. The flow cytometry panel used for this study was conservative and we relied on a simpler approach using an antibody panel that allowed us to identify both monocyte and DC subsets in the same staining. We were limited first and foremost by the cell numbers in nasopharyngeal aspirates, and were not able to perform multiple stainings with different antibody panels on cells from the same sample. Therefore, markers to identify more recently described subsets are lacking, like CD5, CD88, CD34, CD163, AXL and SIGLEC6, and that these markers should be included in future studies. Furthermore, we agree that future studies, both our own and those of others in the field, will greatly benefit from additional single cell analysis of nasopharyngeal immune cells, and from generating transcriptomic or epigenetic profiles of these cells. However, it is a limitation that we are currently unable to overcome. Reviewer 2 also hinted at the limitations of the mechanistic insights obtained from our data. We propose that the greater contribution of our study may not be in explaining the “why and how”, but rather in demonstrating a path towards answering these questions.

Reviewer 2 also hinted at the descriptive nature of the data. Due to the limitations on cell numbers in the nasopharyngeal aspirate samples, we were unable to perform functional assays using these cells which would have provided mechanistic insights into the fate of monocyte and DC subsets recruited to the nasopharynx during infection. Additionally, we were unable to apply multivariate analysis to our data due to stratification issue, i.e. sub-groups had too few data points for statistically significant comparisons. We hope that in future studies we and others are able to address such questions in greater detail. Despite limited cell numbers, the methods described here to obtain nasopharyngeal aspirates have provided important insights into the immune responses in the upper airways during respiratory viral infection in other studies we have published (Falck-Jones et al., https://doi.org/10.1172/JCI144734 and Cagigi et al., https://doi.org/10.1172/jci.insight.151463, Havervall et al. https://doi.org/10.1056/nejmc2209651 and Marking et al. *Lancet Infectious Diseases in press*).

As Reviewer 2 (and Reviewer 1 previously) pointed out, the classification of myeloid cells that we chose was to ensure quantification of the three common subsets of DCs and monocytes. Albeit outdated, the gating strategy we used allowed us to identify two key changes in the myeloid compartment during IAV infection. Firstly in blood, despite an overall reduction in PBMCs/mL blood, overall, CD11c+ cells were increased as compared to healthy individuals (Figure 1F, left). Secondly, as we suspected, the higher cell numbers we observed in the nasopharynx were largely due to an increase of CD11c+ “myeloid” cells.

As Reviewer 2 pointed out, it is true that cDC1 with low CD11c expression may be excluded in our initial gating strategy. However, in subsequent gates, while we were able to identify a clearly distinguishable population of CD11c–CD123–CD141+ cells in NPA (Figure 2—figure supplement 1), the frequency of these CD11c^low^ cDC1s was extremely low and unsurprising, as other DC subsets identified in the NPA of healthy individuals have also been very infrequent (marked-up version lines 299-305). During IAV infection, the frequencies were increased in NPA allowing the identification of this subset (CD11c^low^ cDC1s) in a handful of IAV patients. Importantly, the pattern we observed in other DC subsets (i.e. decreased in blood and increased in NPA) was upheld in this instance as well.

The intermediate monocyte is really just an activated classical monocyte. This caveat could be discussed and perhaps some supporting data put in to show an alternative way of analysing the classical monocyte pool, using CD16 as a marker of activation, rather than trying to define a subset from the smear.

Our understanding of the literature, and in our opinion, the prevailing interpretation of the research field is that intermediate monocytes (IMs) are an intermediary stage between the classical CD14+CD16– and “nonclassical” CD14–CD16+ monocytes, rather than activated classical monocytes. However, decisive studies to pinpoint this are difficult to do in humans. Several studies have shown that intermediate monocytes are present in healthy individuals at steady state (Patin et al., https://doi.org/10.1038/s41590-018-0049-7 and Patel et al., https://doi.org/10.1084/jem.20170355), suggesting that they are a proportion of monocytes likely following a sequential transition program to become nonclassical monocytes, and are not merely activated monocytes. Whether a discrete subset or not, cells phenotypically defined as intermediate monocytes (CD14+CD16+) are known to expand during acute viral infections and secrete inflammatory markers (Kwissa et al., https://doi.org/10.1016/j.chom.2014.06.001 and Oshansky et al., https://doi.org/10.1164/rccm.201309-1616oc) suggesting that they are a relevant subset to identify to understand immune responses during infection. In the current study, within the monocyte compartment, we observed a remarkable increase in the frequency of CD14+CD16+ intermediate monocytes. Reviewer 2’s suggestion about looking at CD16 as an expression marker is well taken, and in the revised version of the manuscript we comment on this in the introduction (marked-up version 66-69).

SLAN is a good marker for the 'true' non-classical population but this would mean new experiments.

We agree with Reviewer 2 that SLAN is an excellent marker for inclusion in future panel design. Unfortunately, all nasopharyngeal cells collected have been exhausted and it is not possible to include SLAN in the current study.

DC2 and 3 can be separated by CD14 and CD1c as Jahnsen and colleagues previously showed with their description of CD1c+ 'monocytes' (DC3) 10.1371/journal.pone.0157387. A lot has been written on these recently but it is tricky to split properly without CD163, CD5 or BTLA. The CD123+ population can be displayed against CD11c to identify the space where AXL^+^SIGLEC6+ pre-DC sit (CD11c slightly positive) but again, easier with a more specific pDC antigen or AXL.

We thank Reviewer 2 for their suggestions to identify DC3 and pre-DC. As already mentioned in the response to Reviewer 1, we have identified cells expressing CD14 and CD1c, “mo-DC”, and this data has been added to the revised manuscript (marked-up version lines 203-204, 215-218, 245-257 and 603). Due to the lack of CD163, BTLA or CD5, we hesitate to call this population *bona fide* DC3. The mo-DC population likely comprises both CD88- (likely CD14+ DC3) and CD88+ cells (true monocyte-derived DCs). Despite not having CD163, BTLA or CD5 in the panel, we believe that this data (on mo-DCs) is relevant to the manuscript and have included it in the revised version of the manuscript in new Figure 2H (marked-up version line 245-257). Unfortunately, it is impossible to distinguish pre-DC from pDC using our existing panel due to the lack of AXL.